# Consistency Models for Scalable and Fast Simulation-Based Inference

**Marvin Schmitt**[*]
University of Stuttgart
Germany
mail.marvinschmitt@gmail.com

**Valentin Pratz**[*]
Heidelberg University & ELIZA
Germany

**Ullrich Köthe**
Heidelberg University
Germany

**Paul-Christian Bürkner**
TU Dortmund University
Germany

**Stefan T. Radev**
Rensselaer Polytechnic Institute
United States

## Abstract

Simulation-based inference (SBI) is constantly in search of more expressive and efficient algorithms to accurately infer the parameters of complex simulation models. In line with this goal, we present consistency models for posterior estimation (CMPE), a new conditional sampler for SBI that inherits the advantages of recent unconstrained architectures and overcomes their sampling inefficiency at inference time. CMPE essentially distills a continuous probability flow and enables rapid few-shot inference with an unconstrained architecture that can be flexibly tailored to the structure of the estimation problem. We provide hyperparameters and default architectures that support consistency training over a wide range of different dimensions, including low-dimensional ones which are important in SBI workflows but were previously difficult to tackle even with unconditional consistency models. Our empirical evaluation demonstrates that CMPE not only outperforms current state-of-the-art algorithms on hard low-dimensional benchmarks, but also achieves competitive performance with much faster sampling speed on two realistic estimation problems with high data and/or parameter dimensions.

## 1 Introduction

Simulation-based inference (SBI) comprises a family of computational methods for modeling the hidden properties of complex systems by means of *simulation* [1, 2]. In recent years, deep learning – and generative models in particular – have proven indispensable for scaling up SBI to challenging inverse problems [3–7]. Recently, multiple streams of neural SBI research have been capitalizing on the rapid progress in generative modeling of unstructured data by re-purposing existing generative architectures into general inverse problem solvers for applications in the sciences [8–11]. In line with the above trend, we contribute a new contender to the growing suite of SBI methods for amortized Bayesian inference.

Within the class of deep generative models, score-based diffusion models [12–15] and flow matching algorithms [16, 17] have recently attracted significant attention due to their sublime performance as realistic generators. Diffusion models and flow matching are strikingly flexible but require a relatively expensive multi-step sampling phase to denoise samples [12]. To address this shortcoming, Song et al. [18] proposed *consistency models* (CMs), which are trained to perform few-step generation by design and have since emerged as a standalone generative model family. Thus, CMs appear attractive

---

[*]equal contribution

38th Conference on Neural Information Processing Systems (NeurIPS 2024).

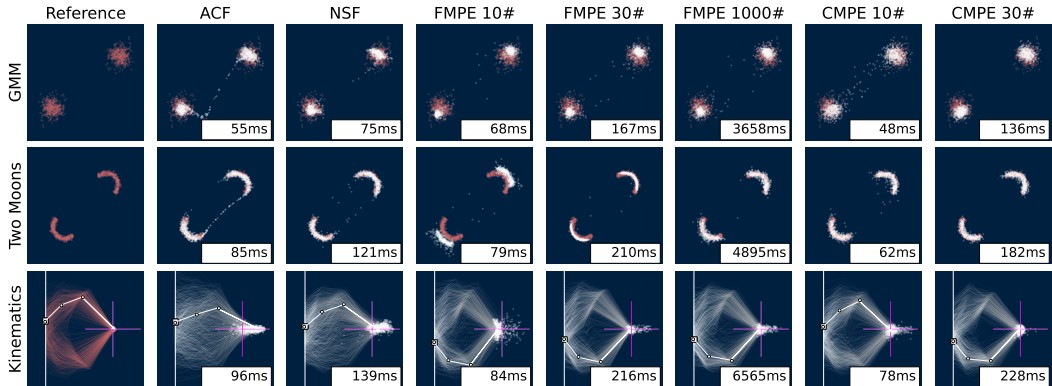

Figure 1: **Experiments 1–3.** 1000 posterior draws for one unseen test instance per task, as well as sampling time in milliseconds. All amortized neural approximators were trained with a small budget of $M = 1024$ simulations. The bottom row shows the posterior predictive distribution in the kinematics task, and the pink cross-hair indicates the true end location $\mathbf{x}$ of the robot arm. Across all benchmarks, CMPE (Ours) yields the best trade-off between fast sampling speed and high accuracy. **ACF:** affine coupling flow, **NSF:** neural spline flow, **FMPE:** flow matching posterior estimation, **CMPE:** consistency model posterior estimation (Ours), **K#** denotes $K$ sampling steps during inference.

as a backbone architecture for SBI workflows due to their unique combination of unconstrained architectures and fast sampling speed – a conceptual advantage over the current state-of-the-art in neural SBI.

However, paradoxically, CMs may struggle with low-dimensional estimation problems, which are less important for image generation but crucial in typical SBI applications: These SBI applications are often characterized by relatively low-dimensional parameter spaces, even though the conditioning vectors (i.e., observables) might be high-dimensional [e.g., 3, 19–21]. Additionally, the quality of *conditional* CMs as few-step Bayesian samplers has not yet been explored empirically (e.g., in terms of probabilistic calibration and precision), even though this is crucial for their application in science and engineering. Lastly, while CMs for image generation are trained on enormous amounts of data, training data are typically scarce in SBI applications. In our empirical evaluations, we demonstrate that CMs are competitive with state-of-the-art SBI algorithms in low-data regimes using our adjusted settings (see Appendix A for details).

In this paper, we propose conditional CMs for SBI along with generalizable hyperparameter settings that achieve competitive performance with much faster sampling speed on both challenging low-dimensional benchmarks and high-dimensional applied problems. Our main contributions are:

1. We adapt consistency models to simulation-based Bayesian inference and propose *consistency model posterior estimation* (CMPE);

2. We showcase the fundamental advantages of consistency models for amortized simulation-based inference: expressive unconstrained architectures *and* fast inference;

3. We demonstrate that CMPE outperforms NPE (normalizing flows) on three benchmark experiments (see Figure 1 above), is competitive with FMPE (flow matching) on Bayesian image denoising, and surpasses both NPE and FMPE on a challenging scientific simulator of tumor spheroid growth.

## 2 Preliminaries and related work

This section recaps simulation-based inference (SBI) via normalizing flows, flow matching, and score-based diffusion models. Readers familiar with these topics can safely fast-forward to Section 3.

## 2.1 Notation

In the following, the neural network training relies on a synthetic *training set* $\{(\boldsymbol{\theta}^{(m)}, \mathbf{x}^{(i)})\}_{m=1}^{M}$, which consists of parameter-data tuples. Each superscript $m$ marks one training example, namely, a tuple of a latent parameter vector and an observable data set which was generated from the parameter vector. The total *simulation budget* for training the neural networks follows as $M$. We summarize the $D$-dimensional latent parameter vector of the simulator as $\boldsymbol{\theta} \equiv (\theta_1, \dots, \theta_D)$. Further, each data set $\mathbf{x} \equiv \{\mathbf{x}_n\}_{n=1}^{N}$ is a matrix whose rows consist of $N$ vector-valued observations. Accordingly, one parameter vector $\boldsymbol{\theta}^{(m)}$ yields one data set $\mathbf{x}^{(m)}$ from the simulator (see Section 2.2). In Bayesian inference, we aim to infer the unknown parameters $\boldsymbol{\theta}$ of the mechanistic simulator, which are not to be confused with the trainable neural network weights $\phi$.

## 2.2 Simulation-based inference (SBI)

Computer simulations play a fundamental role across countless scientific disciplines, ranging from physics to biology, and from climate science to economics [2]. Simulators $g(\cdot, \cdot)$ generate observables $\mathbf{x} \in \mathcal{X}$ as a function of unknown parameters $\boldsymbol{\theta} \in \Theta$ and latent program states $\boldsymbol{\xi} \in \Xi$ [1]:

$$\mathbf{x} = g(\boldsymbol{\theta}, \boldsymbol{\xi}) \quad \text{with } \boldsymbol{\theta} \sim p(\boldsymbol{\theta}), \; \boldsymbol{\xi} \sim p(\boldsymbol{\xi} \,|\, \boldsymbol{\theta}) \tag{1}$$

The forward problem in Equation 1 is typically well-understood through scientific theories instantiated as generative models. The inverse problem, however, is much harder, and forms the crux of Bayesian inference: reduce uncertainty about the unknowns $\boldsymbol{\theta}$ based on observables $\mathbf{x}$ through the *posterior distribution* $p(\boldsymbol{\theta} \,|\, \mathbf{x}) \propto p(\boldsymbol{\theta}) \, p(\mathbf{x} \,|\, \boldsymbol{\theta})$. The key property of neural SBI and early approximate Bayesian computation [ABC; 22, 23] methods is that they approach the inverse problem by sampling (i.e., simulating) from the forward model. Thus, the forward model acts as an *implicit statistical model* [24] that enables proper Bayesian inference in the limit of infinite simulations. In contrast, likelihood-based methods (e.g., MCMC) depend on the explicit evaluation of the likelihood $p(\mathbf{x} \,|\, \boldsymbol{\theta})$.

We can classify neural SBI methods into *sequential* [e.g., 10, 25–29] and *amortized* [e.g., 5, 9–11, 30–35] methods. Sequential inference algorithms iteratively refine the prior $p(\boldsymbol{\theta})$ to generate simulations in the vicinity of the target observation. Thus, they are *not amortized*, as each new observed data set requires a potentially costly re-training of the neural approximator tailored to the particular target observation. However, most sequential methods can be turned *amortized* by training the neural approximator to generalize over the entire prior predictive space of the model. This allows us to query the approximator with any new data set during inference. In fact, amortization can be performed across any component of the model, including multiple data sets [5] and contextual factors, such as the number of observations in a data set [31], heterogeneous data sources [36], or even different probabilistic models [37, 38]. Our current work is situated in the amortized setting.

## 2.3 Normalizing flows for neural posterior estimation

Neural posterior estimation (NPE) methods for SBI have traditionally relied on conditional discrete normalizing flows for learning a conditional neural density estimator $p_{\phi}(\boldsymbol{\theta} \,|\, \mathbf{x})$ from simulated pairs $(\boldsymbol{\theta}, \mathbf{x})$ of parameters and data [8, 39]. Normalizing flows can learn closed-form approximate densities via maximum likelihood training:

$$\mathcal{L}_{\text{NPE}} = \mathbb{E}_{p(\boldsymbol{\theta}, \mathbf{x})}[-\log p_{\phi}(\boldsymbol{\theta} \,|\, \mathbf{x})] \tag{2}$$

Two prominent normalizing flow architectures in SBI are *affine coupling flows* [ACF; 40] and *neural spline flows* [NSF; 41]. A major disadvantage of these architectures is their strict bijectivity requirement: it restricts the design space of the neural networks to invertible functions with cheap Jacobian calculations, hence the desire for more flexible architectures. In the following, we will briefly recap recent works that have pioneered the use of *multi-step, unconstrained* architectures for SBI, inspired by the success of such models in a variety of generative tasks [13, 15].

## 2.4 Flow matching for posterior estimation

Wildberger et al. [11] applied flow matching techniques [16, 17] in SBI, an approach abbreviated as flow matching posterior estimation (FMPE). FMPE is based on optimal transport [42, 43], where the mapping between base and target distribution is parameterized by a continuous process driven

by a vector field $\mu$ on the sample space for each time step $t \in [0, 1]$. Here, the distribution at $t = 1$ could be a unit Gaussian $\mathcal{N}(\mathbf{0}, \mathbf{I})$ and the distribution at $t = 0$ is the target posterior. The FMPE loss replaces the maximum likelihood term in the NPE objective (Eq. 2) with a conditional flow matching objective,

$$\mathcal{L}_{\text{FMPE}} = \mathbb{E}_{p(\mathbf{x}, \boldsymbol{\theta})} \left[ \int_0^1 \| u_t(\boldsymbol{\theta}_t \,|\, \boldsymbol{\theta}) - \mu_{\boldsymbol{\phi}}(\boldsymbol{\theta}_t, t; \mathbf{x}) \|^2 \, \mathrm{d}t \right], \tag{3}$$

where $\mu_{\boldsymbol{\phi}}$ denotes the conditional vector field parameterized by a unconstrained neural network with trainable parameters $\boldsymbol{\phi}$, and $u_t$ denotes a marginal vector field, which can be as simple as $u_t(\boldsymbol{\theta}_t \,|\, \boldsymbol{\theta}) = \boldsymbol{\theta}_1 - \boldsymbol{\theta}$ for all $t$ [17]. We obtain posterior draws $\boldsymbol{\theta}_0 \sim p(\boldsymbol{\theta} \,|\, \mathbf{x})$ by solving $\mathrm{d}\boldsymbol{\theta}_t = -\mu(\boldsymbol{\theta}_t, t; \mathbf{x})$ in reverse on $t \in [0, 1]$, starting with noise samples $\boldsymbol{\theta}_1 \sim \mathcal{N}(\mathbf{0}, \mathbf{I})$. We can use any off-the-shelf ODE solver for transforming noise $\boldsymbol{\theta}_1$ into a draw $\boldsymbol{\theta}_0$ from the approximate posterior. In principle, the number of steps $K$ in the ODE solver can be adjusted by setting the step size $\mathrm{d}t = 1/K$. Decreasing the number of steps increases the sampling speed, but FMPE is not designed to optimize few-step sampling performance. We confirm this in our experiments with a rectified flow that encourages straight-path solutions [17].

## 2.5 Neural posterior score estimation

Another approach to simulation-based inference with unconstrained networks lies in neural posterior score estimation [NPSE; 10], which can either feature single-round inference (amortized) or multiple sequential inference rounds on a particular data set (non-amortized). This stream of research uses conditional score-based diffusion models [12, 44] to learn the posterior distribution. With a slightly different focus, [9] factorize the posterior distribution and learn scores of the diffused posterior for subsets (down to a single observation) of a larger data set. Subsequently, the information from the subsets is aggregated by combining the learned scores to approximate the posterior distribution of the entire data set. Crucially, both methods rely on the basic formulation of score-based diffusion models: They gradually diffuse the target distribution according to the following diffusion process,

$$\mathrm{d}\boldsymbol{\theta}_t = \mu(\boldsymbol{\theta}_t, t; \mathbf{x})\mathrm{d}t + \sigma(t)\mathrm{d}\mathbf{w}_t, \tag{4}$$

with drift coefficient $\mu$, diffusion coefficient $\sigma$, time $t \in [0, T]$, and Brownian motion $\{\mathbf{w}_t\}_{t \in [0, T]}$. At each time $t$, the current (diffused) distribution of $\boldsymbol{\theta}$ conditional on $\mathbf{x}$ is denoted as $p_t(\boldsymbol{\theta} \,|\, \mathbf{x})$. Crucially, the distribution at $t = 0$ equals the target posterior distribution, $p_0(\boldsymbol{\theta} \,|\, \mathbf{x}) \equiv p(\boldsymbol{\theta} \,|\, \mathbf{x})$, and the distribution at $t = T$ is set to be Gaussian noise (see below). Song et al. [12] prove that there exists an ordinary differential equation ("Probability Flow ODE") whose solution trajectories at time $t$ are distributed according to $p_t(\boldsymbol{\theta} \,|\, \mathbf{x})$,

$$\mathrm{d}\boldsymbol{\theta}_t = \left[ \mu(\boldsymbol{\theta}_t, t; \mathbf{x}) - \frac{1}{2}\sigma(t)^2 \, \nabla \log p_t(\boldsymbol{\theta}_t \,|\, \mathbf{x}) \right]\mathrm{d}t, \tag{5}$$

where $\nabla \log p_t(\boldsymbol{\theta}_t \,|\, \mathbf{x})$ is the score function of $p_t(\boldsymbol{\theta} \,|\, \mathbf{x})$. This differential equation is usually designed to yield a spherical Gaussian noise distribution $p_T(\boldsymbol{\theta} \,|\, \mathbf{x}) \approx \mathcal{N}(\mathbf{0}, T^2\mathbf{I})$ after the diffusion process. Since we do not have access to the target posterior $p(\boldsymbol{\theta} \,|\, \mathbf{x})$, score-based diffusion models train a time-dependent score network $s_{\boldsymbol{\phi}}(\boldsymbol{\theta}_t, t, \mathbf{x}) \approx \nabla \log p_t(\boldsymbol{\theta}_t \,|\, \mathbf{x})$ via score matching and insert it into Eq. 5. Setting $\mu(\boldsymbol{\theta}_t, t; \mathbf{x}) = 0$ and $\sigma_t = \sqrt{2t}$, the estimate of the Probability Flow ODE becomes $\mathrm{d}\boldsymbol{\theta}_t = -ts_{\boldsymbol{\phi}}(\boldsymbol{\theta}_t, \mathbf{x}, t)\mathrm{d}t$ [45]. Finally, we can generate a random draw from the noise distribution $\boldsymbol{\theta}_T \sim \mathcal{N}(\mathbf{0}, T^2\mathbf{I})$ and solve the Probability Flow ODE backwards for a trajectory $\{\boldsymbol{\theta}_t\}_{t \in [T, 0]}$. The end of the trajectory $\boldsymbol{\theta}_0$ represents a draw from the approximate posterior $p_0(\boldsymbol{\theta}_0 \,|\, \mathbf{x}) \approx p(\boldsymbol{\theta} \,|\, \mathbf{x})$.

**Numerical stability.** The solver is usually stopped at a fixed small positive number $t = \varepsilon$ to prevent numerical instabilities [18], so we use $\boldsymbol{\theta}_\varepsilon$ to denote the draw from the approximate posterior. For simplicity, we will also refer to $\boldsymbol{\theta}_\varepsilon$ as the *trajectory's origin*.

## 3  Consistency model posterior estimation

Diffusion models have one crucial drawback: At inference time, they require solving a differential equation for each posterior draw which slows down their sampling speed. This is particularly troublesome in SBI applications which may require thousands of samples for thousands of data sets [e.g., >1M data sets in 7]. Consistency models [CMs; 18] address this problem with a new generative model family that supports both single-step and multi-step sampling. In the following, we summarize key ideas of CMs [18, 46] with a focus on conditional sampling for the purpose of SBI.

## 3.1 Conditional consistency models

The consistency function $f : (\boldsymbol{\theta}_t, t; \mathbf{x}) \mapsto \boldsymbol{\theta}_\varepsilon$ maps points across the solution trajectory $\{\boldsymbol{\theta}_t\}_{t \in [T, \varepsilon]}$ to the trajectory's origin $\boldsymbol{\theta}_\varepsilon$ given a fixed conditioning variable (i.e., observation) $\mathbf{x}$ and the probability flow ODE in Eq. 5. To achieve this with established score-based diffusion model architectures, we can use a unconstrained neural network $F_\phi(\boldsymbol{\theta}, t; \mathbf{x})$ which is parameterized through skip connections of the form

$$f_\phi(\boldsymbol{\theta}, t; \mathbf{x}) = c_{\text{skip}}(t)\boldsymbol{\theta} + c_{\text{out}}(t)F_\phi(\boldsymbol{\theta}, t; \mathbf{x}), \tag{6}$$

where $c_{\text{skip}}(t)$ and $c_{\text{out}}(t)$ are differentiable and fulfill the boundary conditions $c_{\text{skip}}(\varepsilon) = 1$ and $c_{\text{out}}(\varepsilon) = 0$. Consistency models are originally motivated as a distillation technique for diffusion models. However, Song et al. [18] show that training consistency models in isolation is possible, and we base our method on their direct approach.

**Sampling.** Once the consistency model has been trained, generating draws from the approximate posterior is straightforward by drawing samples from the noise distribution, $\boldsymbol{\theta}_T \sim \mathcal{N}(\mathbf{0}, T^2\mathbf{I})$, which shall then be transformed into samples from the target distribution, like in a standard diffusion model. In contrast to diffusion models, however, we do not need to solve a sequence of differential equations for this transformation. Instead, we can use the learned consistency function $f_\phi$ to obtain the one-step target sample $\boldsymbol{\theta}_\varepsilon = f_\phi(\boldsymbol{\theta}_T, T; \mathbf{x})$. What is more, inference with consistency models is not actually limited to one-step sampling. In fact, multi-step generation is possible with an iterative sampling procedure, which we will describe in the following. For a sequence of time points $\varepsilon = t_1 < t_2 < \cdots < t_K = T$ and initial noise $\boldsymbol{\theta}_K \sim \mathcal{N}(\mathbf{0}, T^2\mathbf{I})$, we calculate

$$\boldsymbol{\theta}_k \leftarrow f_\phi(\boldsymbol{\theta}_{k+1}, t_{k+1}; \mathbf{x}) + \sqrt{t_k^2 - \varepsilon^2}\mathbf{z}_k \tag{7}$$

for $k = K - 1, K - 2, \ldots, 1$, where $\mathbf{z}_k \sim \mathcal{N}(\mathbf{0}, \mathbf{I})$ and $K - 1$ is the number of sampling steps [18, 46]. The resulting sample is usually better than a one-step sample.

## 3.2 Consistency models for simulation-based inference

Originally developed for image generation, consistency models can be applied to learn arbitrary distributions. The unconstrained architecture enables the integration of specialized architectures for both the data $\mathbf{x}$ and the parameters $\boldsymbol{\theta}$. Due to the low number of passes required for sampling (in contrast to flow matching and diffusion models), more complex networks can be used while maintaining low inference time. In theory, consistency models combine the best of both worlds: unconstrained networks for optimal adaptation to parameter structure and data modalities, while enabling fast inference speed with few network passes. Currently, this comes at the cost of explicit invertibility, which limits the computation of posterior densities. More precisely, single-step consistency models do not allow density evaluations at an arbitrary parameter value $\boldsymbol{\theta}$ but only at a set $S$ of approximate posterior draws $\{\boldsymbol{\theta}_\varepsilon^{(1)}, \ldots, \boldsymbol{\theta}_\varepsilon^{(S)}\}$. However, this is sufficient for important downstream tasks like marginal likelihood estimation, importance sampling, or self-consistency losses. In contrast, multi-step consistency sampling defines a Markov chain which cannot be evaluated without an additional density estimator (see Section 3.5 for details). In accordance with the taxonomy from Cranmer et al. [1], we call our method *consistency model posterior estimation* (CMPE). While we focus on posterior estimation, using consistency models for likelihood emulation is a natural extension of our work.

As a consequence of its fundamentally different training objective, CMPE is not just a faster version of FMPE. Instead, it shows qualitative differences to FMPE beyond a much faster sampling speed. In **Experiment 4**, we show that CMPE is less dependent on the neural network architecture than FMPE, making it a promising alternative when the optimal architecture for a task is not known. Further, we consistently observe good performance in the low-data regime, rendering CMPE an attractive method when training data is scarce. In fact, data availability is a common limiting factor for complex simulators in engineering [47] and science [e.g., molecular dynamics; 48].

## 3.3 Optimization objective

We formulate the consistency training objective for CMPE, which extends the unconditional training objective from Song and Dhariwal [46] with a conditioning variable $\mathbf{x}$ to cater to the SBI setting,

$$\mathcal{L}_{\text{CMPE}}(\phi, \phi^-) = \mathbb{E}\Big[\lambda(t_i)\, d(\mathbf{u}(\phi, t_{i+1}; \mathbf{x}), \mathbf{u}(\phi^-, t_i; \mathbf{x}))\Big] \tag{8}$$

where $\lambda(t)$ is a weighting function, $d(\mathbf{u}, \mathbf{v})$ is a distance metric, $\mathbf{z} \sim \mathcal{N}(\mathbf{0}, \mathbf{I})$ is unit Gaussian noise, and the arguments for the distance metric amount to the outputs of the consistency function obtained at two neighboring time indices $t_i$ and $t_{i+1}$,

$$\mathbf{u}(\boldsymbol{\phi}, t_{i+1}; \mathbf{x}) = f_{\boldsymbol{\phi}}(\boldsymbol{\theta} + t_{i+1}\mathbf{z}, t_{i+1}; \mathbf{x}), \quad \mathbf{u}(\boldsymbol{\phi}^-, t_i; \mathbf{x}) = f_{\boldsymbol{\phi}^-}(\boldsymbol{\theta} + t_i\mathbf{z}, t_i; \mathbf{x}). \tag{9}$$

The teacher's neural network weights $\boldsymbol{\phi}^-$ are a copy of the student's weights which are held constant during each step via a `stopgrad` operator, $\boldsymbol{\phi}^- \leftarrow \texttt{stopgrad}(\boldsymbol{\phi})$. We follow Song and Dhariwal [46] using $\lambda(t_i) = 1/(t_{i+1} - t_i)$ and $d(\mathbf{u}, \mathbf{v}) = \sqrt{\|\mathbf{u} - \mathbf{v}\|_2^2 + c^2} - c$ from the Pseudo-Huber metric family. Appendix A contains more details on discretization and noise schedules.

### 3.4 Hyperparameter tuning

Consistency training introduces several additional hyperparameters, but there is limited theoretical guidance on how to select most of them. Within the scope of this paper, we rely on empirical search to identify the hyperparameters that are the most relevant for tuning efforts. As stated above, the maximum time $T$ determines the standard deviation of the latent distribution. It should be larger than the magnitude of the target distribution [18]. Values that are too low might lead to biased sampling, while values that are too high can make the training process more difficult. The minimum and maximum number of discretization steps $s_0, s_1$ during training also influence the final result in our experiments. For $s_0$, a sufficiently low value (e.g. 10) should be chosen. For $s_1$, smaller values around 50 seem beneficial for stable training, especially when the number of epochs is low. When longer training is possible, increasing $s_1$ might lead to improved accuracy. We summarize our recommended hyperparameter choices for SBI applications in Appendix C.

### 3.5 Density estimation

Using the change-of-variable formula, we can express the posterior density of a single-step sample as

$$p_\varepsilon(\boldsymbol{\theta} \mid \mathbf{x}) = p_T(\boldsymbol{\theta}_T = f_{\boldsymbol{\phi}}^{-1}(\boldsymbol{\theta}_\varepsilon, T; \mathbf{x})) \left| \det \left( \frac{\partial \boldsymbol{\theta}_T}{\partial \boldsymbol{\theta}_\varepsilon} \right) \right|, \tag{10}$$

where $f_{\boldsymbol{\phi}}^{-1}(\boldsymbol{\theta}, T; \mathbf{x})$ is the *implicit* inverse of the consistency function and $\partial \boldsymbol{\theta}_T / \partial \boldsymbol{\theta}_\varepsilon$ is the resulting Jacobian. While we cannot explicitly evaluate $f_{\boldsymbol{\phi}}^{-1}(\boldsymbol{\theta}_\varepsilon, T; \mathbf{x})$, we can generate samples from $\boldsymbol{\theta}_T \sim \mathcal{N}(\mathbf{0}, T^2\mathbf{I})$ and use autodiff for evaluating the Jacobian because the consistency surrogate $f_{\boldsymbol{\phi}}$ is differentiable. Thus, we cannot directly evaluate the posterior density at *arbitrary* $\boldsymbol{\theta}$. Yet, evaluating the density at a set of $S$ posterior draws $\{\boldsymbol{\theta}_\varepsilon\}_{s=1}^S$ suffices for crucial downstream tasks, such as marginal likelihood estimation [49], neural importance sampling [21], or self-consistency losses [50].

For the purpose of *multi-step sampling*, the latent variables $\boldsymbol{\theta}_K, \boldsymbol{\theta}_{K-1}, \ldots, \boldsymbol{\theta}_1$ form a Markov chain with a change-of-variables (CoV) given by Köthe [51]:

$$p(\boldsymbol{\theta}_1 \mid \mathbf{x}) = p(\boldsymbol{\theta}_K) \prod_{k=2}^K \frac{p(\boldsymbol{\theta}_{k-1} \mid \boldsymbol{\theta}_k, \mathbf{x})}{p(\boldsymbol{\theta}_k \mid \boldsymbol{\theta}_{k-1}, \mathbf{x})}. \tag{11}$$

Due to Eq. 7, the backward conditionals are given by $p(\boldsymbol{\theta}_{k-1} \mid \boldsymbol{\theta}_k, \mathbf{x}) = \mathcal{N}(f_{\boldsymbol{\phi}}(\boldsymbol{\theta}_k, t_k; \mathbf{x}), (t_k^2 - \varepsilon^2)\mathbf{I})$. Unfortunately, the forward conditionals $p(\boldsymbol{\theta}_k \mid \boldsymbol{\theta}_{k-1}, \mathbf{x})$ are not available in closed form, but we could learn an amortized surrogate model $q(\boldsymbol{\theta}_k \mid \boldsymbol{\theta}_{k-1})$ capable of single-shot density estimation based on a data set of execution paths $\{\boldsymbol{\theta}_{1:K}\}$. This method would work well for relatively low-dimensional $\boldsymbol{\theta}$, which are common throughout scientific applications, but may diminish the efficiency gains of CMPE in higher dimensions. We leave an extensive evaluation of different surrogate density models to future work.

### 3.6 Choosing the number of sampling steps

The design of consistency models enables one-step sampling (see Eq. 7). In practice, however, using two steps significantly increases the quality in image generation tasks [46]. We observe that few-step sampling with approximately $K = 5-15$ steps provides the best trade-off between sample quality and compute for SBI, particularly in low-dimensional problems. This is roughly comparable to the speed of neural one-step estimators like affine coupling flows or neural spline flows in **Experiments**

**1–3** (see Figure 2a). The number of steps can be chosen at inference time, so practitioners can easily adjust this for a given situation. We noticed that approaching the maximum trained number of discretization steps can lead to overconfident posterior distributions. See Section C.5 for an empirical evaluation of the relation between sampling quality and number of sampling steps.

## 4 Empirical evaluation

Our experiments cover three fundamental aspects of SBI. First, we perform an extensive evaluation on three low-dimensional experiments with bimodal posterior distributions from benchmarking suites for inverse problems [52, 53]. The simulation-based training phase is based on a fixed training set $\{(\mathbf{x}^{(m)}, \boldsymbol{\theta}_*^{(m)})\}_{m=1}^M$ of $M$ tuples of data sets $\mathbf{x}^{(m)}$ and corresponding ground-truth parameters $\boldsymbol{\theta}_*^{(m)}$. Second, we focus on an image denoising example which serves as a sufficiently high-dimensional case study in the context of SBI [32, 54]. Third, we apply our CMPE method to a computationally challenging scientific model of tumor spheroid growth and showcase its superior performance for a complex simulator from cell biology [19]. We implement all experiments using the BayesFlow Python library for amortized Bayesian workflows [55].

**Evaluation metrics.** We evaluate the experiments based on well-established metrics to gauge the accuracy and calibration of the results. All metrics are computed on a test set of $J$ unseen instances $\{(\mathbf{x}^{(j)}, \boldsymbol{\theta}_*^{(j)})\}_{j=1}^J$. In the following, $S$ denotes the number of (approximate) posterior samples that we draw for each instance $J$. First, the easy-to-understand root mean squared error (RMSE) metric quantifies both the bias and variance of the approximate posterior samples across the test set as $\frac{1}{J} \sum_{j=1}^J \sqrt{\frac{1}{S} \sum_{s=1}^S \left( \boldsymbol{\theta}_s^{(j)} - \boldsymbol{\theta}_*^{(j)} \right)^2}$. Second, we estimate the squared maximum mean discrepancy [MMD; 56] between samples from the approximate vs. reference posterior, which is a kernel-based distance between distributions. Third, as a widely applied metric in SBI, the C2ST score uses an MLP classifier to distinguish samples from the approximate and reference posteriors. The resulting test accuracy of the classifier is the C2ST score, which ranges from 1 (samples are perfectly separable $\rightarrow$ bad posterior) to 0.5 (samples are indistinguishable $\rightarrow$ perfect posterior). Finally, we assess uncertainty calibration through simulation-based calibration (SBC; [57]): All uncertainty intervals $U_q(\boldsymbol{\theta} \mid \mathbf{x})$ of the true posterior $p(\boldsymbol{\theta} \mid \mathbf{x})$ are well calibrated for every quantile $q \in (0, 1)$,

$$q = \iint \mathbf{I}[\boldsymbol{\theta}_* \in U_q(\boldsymbol{\theta} \mid \mathbf{x})] \, p(\mathbf{x} \mid \boldsymbol{\theta}_*) \, p(\boldsymbol{\theta}_*) \mathrm{d}\boldsymbol{\theta}_* \mathrm{d}\mathbf{x}, \tag{12}$$

where $\mathbf{I}[\cdot]$ is the indicator function. Discrepancies from Eq. 12 indicate deficient calibration of an approximate posterior. The expected calibration error (ECE) aggregates the median SBC error of central credible intervals of 20 linearly spaced quantiles $q$, averaged across the test set.

**Contender methods.** We compare affine coupling flows [ACF; 40], neural spline flows [NSF; 41], flow matching posterior estimation [FMPE; 11], and consistency model posterior estimation (CMPE; ours). We use identical unconstrained neural networks for FMPE and CMPE to ensure comparability. Appendix C lists the hyperparameters of all models in the experiments.

### 4.1 Experiment 1: Gaussian mixture model

We illustrate CMPE on a 2-dimensional Gaussian mixture model with two symmetrical components [9, 50]. The symmetrical components are equally weighted and have equal variance,

$$\boldsymbol{\theta} \sim \mathcal{N}(\boldsymbol{\theta} \mid \mathbf{0}, \mathbf{I}), \ \mathbf{x} \sim \tfrac{1}{2} \mathcal{N}(\mathbf{x} \mid \boldsymbol{\theta}, \tfrac{\mathbf{I}}{2}) + \tfrac{1}{2} \mathcal{N}(\mathbf{x} \mid -\boldsymbol{\theta}, \tfrac{\mathbf{I}}{2}), \tag{13}$$

where $\mathcal{N}(\cdot \mid \mu, \boldsymbol{\Sigma})$ is a Gaussian distribution with location $\mu$ and covariance matrix $\boldsymbol{\Sigma}$. The resulting posterior distribution is bimodal and point-symmetric around the origin (see Figure 1 top-left). Each simulated data set $\mathbf{x}$ consists of ten exchangeable observations $\{\mathbf{x}_1, \ldots, \mathbf{x}_{10}\}$, and we train all architectures on $M = 1024$ training simulations. We use a DeepSet [58] to learn 6 summary statistics for each data set jointly with the inference task.

**Results.** As displayed in Figure 1, both ACF and NSF fail to fit the bimodal posterior with separated modes. FMPE successfully forms disconnected posterior modes, but shows visible overconfidence through overly narrow posteriors. The visual sampling performance of CMPE is superior to all other methods. In this task, we observe that CMPE does not force us to choose between speed *or* performance relative to the other approximators. Instead, CMPE can outperform all other methods

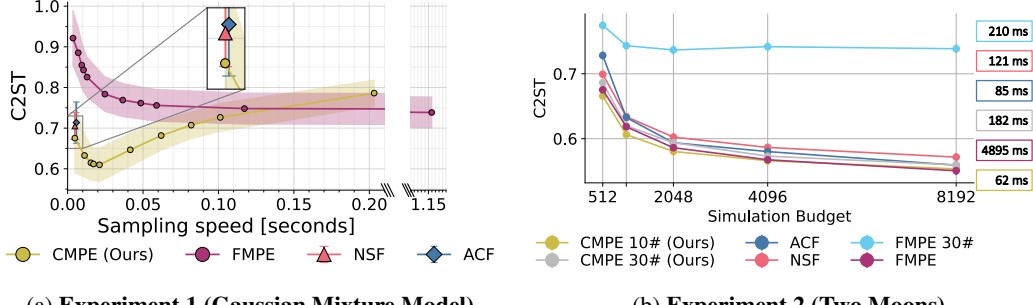

(a) **Experiment 1 (Gaussian Mixture Model)**    (b) **Experiment 2 (Two Moons)**

Figure 2: C2ST score of 4000 approximate posterior draws vs. reference posterior (lower is better) for $J = 100$ unseen test examples. (a) CMPE (Ours) outperforms all other methods through both faster and more accurate inference on the GMM benchmark (mean±SD). (b) CMPE (Ours) with 10 sampling steps shows superior performance up to a training budget of 4096 instances on the Two Moons benchmark (mean±SE).

simultaneously with respect to both speed *and* performance, as quantified by lower C2ST to the reference posterior across $J = 100$ test instances (see Figure 2a). If we tolerate slower sampling, CMPE achieves peak performance at $K = 10$ inference steps. Most notably, CMPE outperforms 1000-step FMPE by a large margin, even though the latter is approximately $75\times$ slower.

## 4.2    Experiment 2: Two moons

This experiment studies the two moons benchmark [27, 49, 52, 59]. The model is characterized by a bimodal posterior with two separated crescent moons for the observed point $\mathbf{x}_{\text{obs}} = (0, 0)^\top$ which an approximator needs to recover. We repeat the experiment for different training budgets $M \in \{512, 1024, 2048, 4096, 8192\}$ to probe each method under varying data availability. While $M = 512$ is a very small budget for the two moons benchmark, $M = 8192$ is considered sufficient.

**Results.** Trained on a simulation budget of $M = 1024$ examples, CMPE consistently explores both crescent moons and successfully captures the local patterns of the posterior (see Figure 1, middle row). Both the affine coupling flow and the neural spline flow fail to fully separate the modes. Most notably, if we aim to achieve fast sampling speed with FMPE by reducing the number of sampling steps during inference, FMPE shows visible overconfidence with 30 sampling steps and markedly deficient approximate posteriors with 10 sampling steps. In stark contrast, CMPE excels in the few-step regime. Figure 2b illustrates that all architectures benefit from a larger training budget. CMPE with 10 steps emerges as the superior architecture in the low- and medium-data regime up to $M = 4096$ training instances. FMPE with 1000 steps outperforms the other approximators by a small margin for the largest training budget of $M = 8192$ instances. Keep in mind, however, that 1000-step FMPE is approximately 30–70× slower than ACF, NSF, and CMPE in this task.

## 4.3    Experiment 3: Inverse kinematics

Proposed as a benchmark task for inverse problems by Kruse et al. [53], the inverse kinematics model aims to reconstruct the configuration $(\theta_1, \theta_2, \theta_3, \theta_4) = \boldsymbol{\theta} \in \mathbb{R}^4$ of a multi-jointed 2D robot arm for a given end position $\mathbf{x} \in \mathbb{R}^2$ (see Figure 1, bottom left). The input to the forward process $g : \boldsymbol{\theta} \mapsto \mathbf{x}$ are the initial height $\theta_1$ of the arm's base, as well as the angles $\theta_2, \theta_3, \theta_4$ at its three joints. The inverse problem aims to determine the posterior distribution $p(\boldsymbol{\theta} \mid \mathbf{x})$, which represents all arm configurations $\boldsymbol{\theta}$ that end at the observed 2D position $\mathbf{x}$ of the arm's end effector.

**Results.** On the challenging small training budget of $M = 1024$ training examples, CMPE with $K = 30$ sampling steps visually outperforms all other methods with respect to posterior predictive performance while maintaining fast inference (see Figure 1). On the C2ST metric, CMPE outperforms normalizing flows (ACF, NSF), but is in turn surpassed by 1000-step FMPE (see Figure 4b), which is however 85× slower (FMPE: 6565ms, CMPE: 78ms). In contrast to the other empirical evaluations, the simulator in this benchmark lacks aleatoric uncertainty [60] and the plausible parameter values are tightly bounded through the geometry of the problem, which might explain this pattern.

## 4.4 Experiment 4: Bayesian denoising

This experiment demonstrates the feasibility of CMPE for a high-dimensional inverse problem, namely, Bayesian denoising on the Fashion MNIST data set [61]. The unknown parameter $\boldsymbol{\theta} \in \mathbb{R}^{784}$ is the flattened original image, and the observation $\mathbf{x} \in \mathbb{R}^{784}$ is a blurred and flattened version of the crisp image from a simulated noisy camera [32, 49, 54]. We compare CMPE and FMPE in a standard (60k training images) and a small data (2k training images) regime. As both methods allow for unconstrained architectures, we can assess the effect of the architectural choices on the results by evaluating both methods using a suboptimal naïve architecture vs. the established U-Net [62].

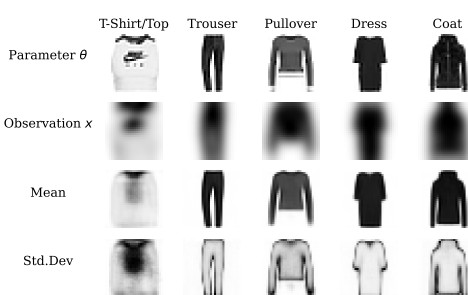

Figure 3: **Experiment 4**. CMPE denoising results on Fashion MNIST (U-Net backbone, $K = 2$ sampling steps, 60 000 training images). *First row:* Original image (target parameters $\boldsymbol{\theta}$). *Second row:* Blurred image (observations $\mathbf{x}$). *Third and fourth row:* Means and standard deviations of the approximate posteriors. *Note:* For standard deviations, darker regions indicate larger variability in the outputs. Adapted from [49]. More in Appendix C.6.

**Neural architectures.** The naïve architecture consists of a convolutional neural network [CNN; 63] to convert the observation into a vector of latent summary statistics. We concatenate input vector, summary statistics, and a time embedding and feed them into a multi-layer perceptron (MLP) with four hidden layers consisting of 2048 units each. This corresponds to a situation where the structure of the observation (i.e., image data) is known, but the structure of the parameters is unknown or does not inform a bespoke network architecture. In this example, however, we can leverage the prior knowledge that our parameters are images. Specifically, we can incorporate inductive biases into our network architecture by choosing a U-Net architecture which is optimized for image processing [i.e., an adapted version of 64]. Again, a CNN learns a summary vector of the noisy observation, which is then concatenated with a time embedding into the conditioning vector for the neural density estimator.

**Results.** We report the aggregated RMSE, MMD, and the time per sample for both methods and architectures (Table 1). FMPE is not able to generate good samples for the naïve architecture, whereas CMPE produces acceptable samples even in this suboptimal setup. This reduced susceptibility to suboptimal architectures might become a valuable feature in high-dimensional problems where the structure of the parameters cannot be exploited. The U-Net architecture enables good sample quality for both

Table 1: **Experiment 4**: RMSE and MMD between the ground-truth image vs. 100 draws from the approximators trained on 2 000 and 60 000 training images, aggregated over 100 test images. For MMD, we draw one denoised sample per test image and calculate the MMD between the denoised samples and the original images. *Time* per draw.

| Models | | RMSE ↓ | | MMD (± SD) [× $10^{-3}$] ↓ | | Time ↓ |
|---|---|---|---|---|---|---|
| | | 2 000 | 60 000 | 2 000 | 60 000 | |
| naïve | FMPE | 0.836 | 0.597 | 171.53 ±1.61 | 95.26 ± 1.21 | 15.4ms |
| | CMPE (Ours) | **0.388** | **0.293** | **102.09** ± 3.24 | **57.90** ± 1.59 | **0.3**ms |
| U-Net | FMPE | **0.278** | **0.217** | **17.38** ± 0.10 | **14.50** ± 0.05 | 565.8ms |
| | CMPE (Ours) | 0.311 | 0.238 | 18.49 ± 0.12 | 16.08 ± 0.05 | **0.5**ms |

methods, highlighting the advantages of unconstrained architectures. The MMD values align well with a visual assessment of the sample quality, therefore we deem it an informative metric to compare the results (see Section C.6 for visual inspection). The U-Net architecture paired with a large training set provides detailed and versatile samples of similar quality for CMPE and FMPE. CMPE enables $50-1000\times$ faster inference than FMPE because it only requires two neural network passes for sampling, while achieving better (naïve) or competitive (U-Net) quality.

## 4.5 Experiment 5: Tumor spheroid growth

We conclude our empirical evaluation with a complex multi-scale model of 2D tumor spheroid growth [19, 65]. The model has 7 unknown parameters which govern single-cell behavior (agent-based) as well as the extracellular matrix description (PDE-based). Crucially, running simulations from this model on consumer-grade hardware is rather expensive ($\approx 1$ minute for a single simulation), so there is a desire for methods that can provide reasonable estimates for a limited offline training budget.

Here, we compare the performance of the four contender methods in this paper on a fixed training set of $M = 19\,600$ simulations with $J = 400$ simulations as a test set to compute performance metrics. The contender methods are affine coupling flows (ACF), neural spline flows (NSF), and flow-matching posterior estimation (FMPE), as these are among the most popular choices for neural SBI to date. Our main contender is FMPE as a free-form architecture, and we use the exact same inference network for both FMPE ($\mu_\phi$ – field network) and CM ($f_\phi$ – consistency network). Across all methods, we use a hybrid LSTM-Transformer architecture to transform high-dimensional summary statistics of variable length into fixed length embedding vectors $h(\mathbf{x})$. Appendix C.7 provides more details on the neural network architectures and training hyperparameters.

**Results.** CMPE outperforms the alternative neural methods via better accuracy and calibration, as indexed by lower RMSE and ECE on 400 unseen test instances (see Table 2). The speed of the simpler ACF is unmatched by the other methods. In direct comparison to its unconstrained contender FMPE, CMPE simultaneously exhibits (i) a slightly higher accuracy; (ii) a drastically improved calibration; and (iii) much faster sampling (see Figure 6 in the Appendix). For this simulator, FMPE did not achieve satisfactory calibration performance with up to $K = 100$ inference steps, so Table 2 reports the best FMPE results for $K = 1000$ steps.

Table 2: **Experiment 5.** RMSE, ECE, and sampling time are computed for 2000 posterior samples, aggregated over 400 unseen test instances. Max ECE denotes the worst-case marginal calibration error across all 7 model parameters.

| Model | RMSE ↓ | Max ECE ↓ | Time ↓ |
|---|---|---|---|
| ACF | 0.589 | 0.021 | **1.07s** |
| NSF | 0.590 | 0.027 | 1.95s |
| FMPE 30# | 0.582 | 0.222 | 17.13s |
| FMPE 1000# | 0.583 | 0.057 | 500.90s |
| CMPE 2# (Ours) | 0.616 | 0.064 | 2.16s |
| CMPE 30# (Ours) | **0.577** | **0.018** | 18.33s |

# 5 Discussion

We presented consistency model posterior estimation (CMPE), a novel approach to perform accurate simulation-based Bayesian inference on large-scale models while achieving fast inference speed. CMPE enhances the capabilities of state-of-the-art neural posterior estimation by combining few-step sampling with unconstrained neural architectures. To assess the effectiveness of CMPE, we applied it to a set of 3 low-dimensional benchmark tasks that allow intuitive visual inspection, as well as a high-dimensional Bayesian denoising experiment and a scientific tumor growth model. Across our experiments, CMPE emerges as a competitive method for simulation-based Bayesian inference, as evidenced by a holistic assessment of posterior accuracy, calibration, and inference speed.

**Limitations.** Our proposed CMPE method has two core limitations, which we specifically highlight again in this paragraph. First, consistency models do not directly yield tractable densities for arbitrary parameter values (see Section 3.5 for a remedy). Second, the relation between inference time (number of sampling steps $K$) and performance (posterior accuracy) is not monotonic, as elucidated in **Experiment 1**, (see Figure 2a). We provide more details and report the CMPE posterior quality (quantified by C2ST) as a function of the number of sampling steps for all three benchmark experiments in Section C.5. This phenomenon is not a *drawback* per se, but rather a counter-intuitive attribute of consistency models. In practice, it can easily be addressed by performing a brief sweep over the number of sampling steps ($K = 1, \ldots, K_{\max}$) during inference time and choosing an optimal $K$ based on user-defined metrics (e.g., calibration or MMD on a validation set). Given the very fast inference speed of CMPE and the observed U-shaped relation between sampling steps and performance, this approach does not generally come with a relevant computational burden.

**Perspectives.** Future work might aim to further reduce the number of sampling steps towards one-step inference in SBI tasks to achieve even faster sampling speed in time-sensitive applications. This might be achieved via extensive automated hyperparameter optimization, tailored training schemes, or consistency model inference in a compressed latent space. Furthermore, consistency trajectory models [CTMs; 66] are a closely related generative model family, whose conditional formulation might prove useful for SBI as well. Overall, our results demonstrate the potential of CMPE as a novel simulation-based inference tool. Its unique combination of highly expressive unconstrained neural networks with fast sampling speed renders CMPE a new contender for SBI workflows in science and engineering, where both time and performance are essential.

**Acknowledgments**

We thank Lasse Elsemüller for insightful feedback and Maximilian Dax for support with the implementation of the gravitational wave experiment in the Dingo library. MS acknowledges funding from the Cyber Valley Research Fund (grant number: CyVy-RF-2021-16) and the Deutsche Forschungsgemeinschaft (DFG, German Research Foundation) under Germany's Excellence Strategy EXC-2075 - 390740016 (the Stuttgart Cluster of Excellence SimTech). MS additionally thanks the Google Cloud Research Credits program and the European Laboratory for Learning and Intelligent Systems (ELLIS) PhD program for support. VP acknowledges funding by the German Academic Exchange Service (DAAD) and the German Federal Ministry of Education and Research (BMBF) through the funding programme "Konrad Zuse Schools of Excellence in Artificial Intelligence" (ELIZA) and support by the state of Baden-Württemberg through bwHPC. VP acknowledges support by the Bundesministerium fuer Wirtschaft und Klimaschutz (BMWK, German Federal Ministry for Economic Affairs and Climate Action) as part of the German government's 7th energy research program "Innovations for the energy transition" under the 03ETE039I HiBRAIN project (Holistic method of a combined data- and model-based Electrode design supported by artificial intelligence).

**Code**

The software code for the experiments is available in a public GitHub repository:
https://github.com/bayesflow-org/consistency-model-posterior-estimation

Additionally, we provide a backend-agnostic implementation of Consistency Model Posterior Estimation (CMPE) in the open source BayesFlow library for amortized Bayesian workflows [55]. Users can choose between a PyTorch, TensorFlow, or JAX backend. BayesFlow is available as open-source software on GitHub (https://github.com/bayesflow-org/bayesflow/) and PyPI.

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

# A  Consistency training details

Table 3 details all functions and parameters required for training. Except for the choice of parameters $s_0$, $s_1$, $\varepsilon$, $T_{\max}$ and $\sigma_{\text{data}}$ we exactly follow the design choices proposed by Song and Dhariwal [46], which improved our results compared to the choices in Song et al. [18]. Song and Dhariwal [46] showed that that the Exponential Moving Average of the teacher's parameter $\phi^-$ should not be used in Consistency Training, therefore we omitted it in this paper.

Table 3: **Design choices**: Table and values adapted to our notation from Song and Dhariwal [46, Table 1].

| | |
|---|---|
| Loss metric | $d(\mathbf{x}, \mathbf{y}) = \sqrt{\|\mathbf{x} - \mathbf{y}\|_2^2 + c^2} - c$ |
| Discretization scheme | $N(k) = \min(s_0 2^{\lfloor k/K' \rfloor}, s_1) + 1$ where $K' = \lfloor K/(\log_2 \lfloor s_1/s_0 \rfloor + 1) \rfloor$ |
| Noise schedule | $t_i$, where $i \sim p(i)$ and $p(i) \sim \text{erf}\left(\frac{\log(t_{i+1}) - P_{\text{mean}}}{\sqrt{2}P_{\text{std}}}\right) - \text{erf}\left(\frac{\log(t_i) - P_{\text{mean}}}{\sqrt{2}P_{\text{std}}}\right)$ |
| Weighting function | $\lambda(t_i) = 1/(t_{i+1} - t_i)$ |
| Skip connections | $c_{\text{skip}}(t) = \sigma_{\text{data}}^2/((t - \varepsilon)^2 + \sigma_{\text{data}}^2), c_{\text{out}}(t) = \sigma_{\text{data}}(t - \varepsilon)/\sqrt{\sigma_{\text{data}}^2 + t^2}$ |
| Parameters | $s_0 = 10, s_1 = 50$ except where indicated otherwise
$c = 0.00054\sqrt{d}$, where $d$ is the dimensionality of $\theta$
$P_{\text{mean}} = -1.1, P_{\text{std}} = 2.0$ |
| | $k \in 0, \dots, K - 1$, where K is the total training iterations
$t_i = \left(\varepsilon^{1/\rho} + \frac{i-1}{N(k)-1}\left(T_{\max}^{1/\rho} - \varepsilon^{1/\rho}\right)\right)^\rho, \rho = 7, \varepsilon = 0.001, T_{\max} = 200$ |

# B  Evaluation metrics

Especially for multi-modal distributions, some of the standard metrics may behave in unexpected ways. This impedes an interpretation of the results and can lead to misleading conclusions. In this section, we want to highlight the peculiarities of some of the metrics, and highlight where caution is necessary when they are applied to the presented benchmarks.

## B.1  C2ST

As an expressive classifier, C2ST is most sensitive to regions without distribution overlap. In our experiments, this results in a pronounced punishment of overconfident posteriors. For limited training data and distributions without sharp edges (e.g., the Gaussian mixture model), some overconfidence is usually given, as samples from the outer tails are simply not present in the training set. As the accuracy is used as a measure, a smaller number of very improbable samples (e.g., samples in the region between two modes) do not influence the results a lot.

In consequence, C2ST may deviate from the visual judgment in important ways. First, overconfident posteriors look more "precise", but are punished by C2ST. Second, samples between modes are very salient, but may not influence the results of C2ST a lot.

For distributions with sharp edges and more or less uniform density in a given mode, c2st is very capable in judging the overlap of two distributions, which then is a good measure of quality. This applies to the two-moons example, where we can observe the expected behavior of a larger training budget leading to more accurate results (Figure 2b).

## B.2  MMD

By capturing and comparing the moments of two distributions, MMD provides a more holistic view than C2ST. It is also applicable to a wider range of cases, as it generalizes to high dimensions. The exact behavior can be controlled by choosing the kernel function. In our experiments, we observe MMD to be less sensitive for distributions with sharp edges (i.e., two moons), where C2ST seems to provide more reasonable results. We also observe a low sensitivity to samples between modes, leading to unexpected results compared to visual inspection.

### B.3 Summary

Calculating meaningful metrics given only samples from two distributions is a challenging task. Ideally, the metrics would reward similar density and punish samples in the low-density regions of the reference distribution. Especially the latter seems to be hard to achieve. Even when metrics work fine for small deviations from the ideal distribution, the behavior for different forms of larger deviations, as they occur in challenging problems or cases with low training budget, might not be easily grasped or interpreted. That makes it difficult to compare different methods producing qualitatively different deviations (e.g., overconfidence vs. underconfidence) from the reference distribution.

## C   Additional details and results

For neural network training, we used a Mac M1 CPU for **Experiments 1–3**, an NVIDIA V100 GPU for **Experiment 4**, an NVIDIA RTX 4090 GPU for **Experiment 5**, and an NVIDIA H100 GPU for **Experiment 6**. The evaluation scripts were executed on a Mac M1 CPU for **Experiments 1–2**, on an NVIDIA V100 GPU for **Experiments 3–4**, on an NVIDIA RTX 4090 GPU for **Experiment 5**, and on an NVIDIA V100 GPU for **Experiment 6**. The computation time for **Experiments 1–5** is approximately one day, with the evaluations (e.g., reference posteriors, C2ST computations, MMD computations taking up a considerable amount of time. However, our proposed CMPE method is designed to be carried out on consumer-grade hardware for applied analyses on real data sets, which usually requires minutes to hours of training and can be executed on CPUs depending on the application. In contrast, the large-scale analysis in **Experiment 6** requires 2–3 days of training time on high-performance computing infrastructure [see 11, for details on this setting].

### C.1   Experiment 1

**Neural network details.** All architectures use a summary network to learn fixed-length representations of the $i.i.d.$ data $\mathbf{x} = \{\mathbf{x}_1, \ldots, \mathbf{x}_{10}\}$. We implement this via a DeepSet [58] with 6-dimensional output. Both ACF and NSF use 4 coupling layers and train for 200 epochs with a batch size 32. CMPE relies on an MLP with 2 hidden layers of 256 units each, L2 regularization with weight $10^{-4}$, 10% dropout and an initial learning rate of $10^{-4}$. The consistency model is instantiated with the hyperparameters $s_0 = 10, s_1 = 1280, T_{\max} = 1$. Training is based on 2000 epochs with batch size 64. FMPE and CMPE use an identical MLP, and the only difference is the initial learning rate of $10^{-5}$ for FMPE to alleviate instable training dynamics.

### C.2   Experiment 2

**Neural network details.** CMPE uses an MLP with 2 layers of 256 units each, L2 regularization with weight $10^{-5}$, 5% dropout, and an initial learning rate of $5 \cdot 10^{-4}$. The consistency model uses $s_0 = 10, s_1 = 50, T_{\max} = 10$, and it is trained with a batch size of 64 for 5000 epochs. Both ACF and NSF use 6 coupling layers of 128 units each, kernel regularization with weight $\gamma = 10^{-4}$, and train for 200 epochs with a batch size 32. FMPE uses the same settings and training configuration as CMPE.

### C.3   Experiment 3

The three robot arm segments have lengths $0.5, 0.5, 1.0$. The parameters $\boldsymbol{\theta} = (\theta_1, \theta_2, \theta_3, \theta_4)$ follow a Gaussian prior $\boldsymbol{\theta} \sim \mathcal{N}(\mathbf{0}, \sigma^2 \mathbf{I})$ with $\sigma^2 = (\frac{1}{16}, \frac{1}{4}, \frac{1}{4}, \frac{1}{4})$. **Neural network details.** CMPE relies on an MLP with 2 layers of 256 units each, L2 regularization with weight $10^{-5}$, 5% dropout, and an initial learning rate of $5 \times 10^{-4}$. The consistency model uses $s_0 = 10, s_1 = 50$, and trains for 2000 epochs with a batch size of 32. FMPE uses the same MLP and training configuration as CMPE. Both ACF and NSF use 6 coupling layers of 128 units each, kernel regularization with weight $\gamma = 10^{-4}$, and train for 200 epochs with a batch size 32. The reference posterior is obtained using approximate Bayesian Computation (ABC), with $\varepsilon = 0.002$, which corresponds to $1/1000$ of the arm length.

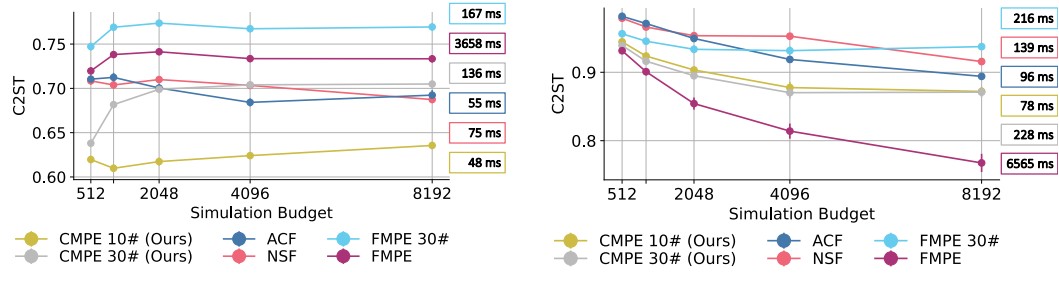

(a) **Experiment 1 (Gaussian Mixture Model)**     (b) **Experiment 3 (Inverse Kinematics)**

Figure 4: C2ST score of 4000 approximate posterior draws vs. reference posterior (lower is better), we report mean±SE over $J = 100$ unseen test examples. (a) For the GMM benchmark, we observe an unexpected pattern for the dependency on the training budget. The C2ST score increases or stays approximately constant across all methods, indicating that in this regime a higher training budget leads to inferior performance, for example due to a tendency to overfit with more training data. It could also be a sign that the C2ST is not a good quality metric for this benchmark, but the monotonically decreasing curve for higher-quality samples (i.e., more sampling steps) for FMPE in Figure 2a indicates that the behavior can probably be attributed to the training budget and not to the metric. See Appendix B for a more detailed discussion. (b) In this task, it is more challenging to achieve excellent C2ST scores because there is no aleatoric uncertainty in the data-generating process. CMPE outperforms ACF and NSF. FMPE performs best and can benefit most from the increased training budget.

## C.4 Unstandardized wall-clock times

[Table 4](#) gives an overview of the observed wall-clock training times for the benchmark tasks in **Experiments 1–3**. These results have been measured on a consumer-grade CPU (Apple Silicon M1 processor). We emphasize that these do not constitute a principled timing benchmark under standardized conditions, and the training times between algorithms are not directly comparable. We train each algorithm for a fixed number of epochs, rather than training until some pre-defined accuracy metric has been achieved. This is because no single reported metric holistically quantifies the accuracy in the investigated benchmark tasks, as explained in more detail in [Appendix B](#). Further, we observed degrading performance of both normalizing flow implementations for larger numbers of epochs despite dropout and regularization. Instead of a comprehensive benchmark timing test, the results shall show that CMPE does in fact require slightly longer training times compared to FMPE given the same neural network backbone. We attribute this to the student-teacher training scheme which necessitates more neural network evaluations (not gradient updates) during training. However, the much faster inference speed of CMPE constitutes a significant advantage for real-time amortized Bayesian inference, which is why the drawback of longer neural network training is acceptable.

| | | $M = 512$ | $M = 1024$ | $M = 2048$ | $M = 4096$ | $M = 8192$ |
|---|---|---|---|---|---|---|
| Gaussian Mixture | ACF | 105 | 189 | 352 | 565 | 1056 |
| | NSF | 118 | 203 | 359 | 646 | 1327 |
| | FMPE | 715 | 1278 | 2008 | 3973 | 7508 |
| | CMPE | 867 | 1471 | 2401 | 4390 | 10102 |
| Two Moons | ACF | 42 | 66 | 118 | 225 | 453 |
| | NSF | 60 | 86 | 146 | 257 | 515 |
| | FMPE | 175 | 289 | 498 | 890 | 1805 |
| | CMPE | 195 | 323 | 571 | 1064 | 2173 |
| Inverse Kinematics | ACF | 43 | 70 | 124 | 234 | 439 |
| | NSF | 63 | 99 | 169 | 297 | 552 |
| | FMPE | 97 | 166 | 311 | 551 | 1095 |
| | CMPE | 107 | 194 | 358 | 719 | 1397 |

Table 4: Wall-clock training times on the benchmark tasks. $M$ denotes the available simulation budget for the neural network training stage.

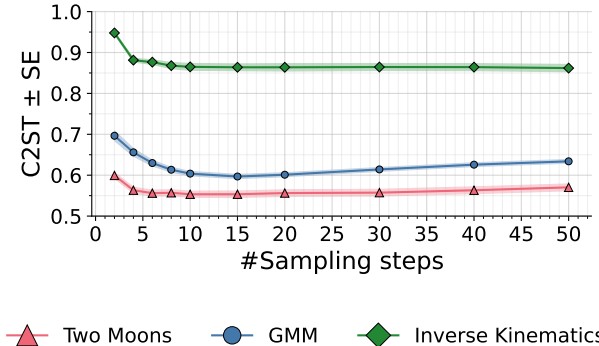

Figure 5: **Experiment 1-3**. C2ST score of 4000 approximate posterior draws vs. reference posterior (lower is better), we report mean±SE over $J = 100$ unseen test examples at different numbers of inference steps. The minimum C2ST value is achieved around 10-20 inference steps for every benchmark, after which the value increases again.

## C.5 Empirical evaluation of optimal number of inference steps

For Experiments 1–3, we evaluate the performance of CMPE (as indexed by C2ST) as a function of the number of sampling steps $K$ during inference. Figure 5 shows that there is no monotonic relationship where more compute leads to better results. Instead, we observe a U-shaped relationship with a performance maximum at $K = 10-20$ sampling steps. This seemingly counter-intuitive phenomenon is a consequence of the consistency training, which aims to achieve few-step sampling and thus has no incentive to optimize the consistency function for a large number $K$ of sampling steps.

## C.6 Experiment 4

The Fashion MNIST data set is released under MIT license [61]. We compare samples from a $2 \times 2 \times 2$ design: Two methods, CMPE and FMPE, are trained on two network architectures, a naïve architecture and a U-Net architecture, using 2 000 and 60 000 training images. Except for the batch size, which is 32 for 2 000 training images and 256 for 60 000 training images, all hyperparameters are held constant between all runs. We use an AdamW optimizer with a learning rate of $5 \cdot 10^{-4}$ and cosine decay for 20 000 iterations, rounded up to the next full epoch. As both methods use the same network sizes, this results in an approximately equal training time between methods (approximately 1-2 hours on a Tesla V100 GPU), with a significantly lower training time for the naïve architecture. **Neural network details.** For CMPE, we use $s_0 = 10, s_1 = 50, \sigma_{\text{data}}^2 = 0.25$ and two-step sampling. For FMPE, we follow Wildberger et al. [11] and use 248 network passes. Note that this is directly correlated with inference time: Choosing a lower value will make FMPE sampling faster, a larger value will slow down FMPE sampling. We show samples for each method (CMPE, FMPE), backbone architecture (MLP, U-Net), and simulation budget (2 000, 6 000) at the end of the appendix.

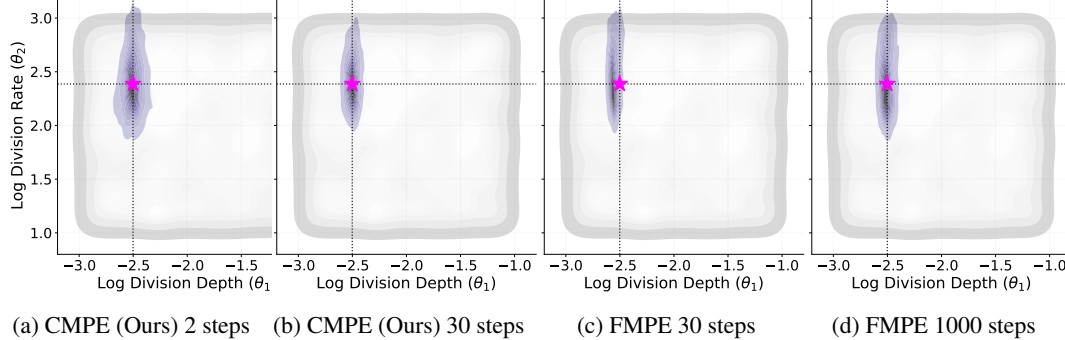

| (a) CMPE (Ours) 2 steps | (b) CMPE (Ours) 30 steps | (c) FMPE 30 steps | (d) FMPE 1000 steps |

Figure 6: **Experiment 5.** Posterior draws for two parameters of the tumor growth model. Cross-hair and pink star indicate the ground-truth parameter that we aim to recover for one fixed data set $\mathbf{x}$. The gray region depicts the prior distribution. CMPE (Ours) shows no visible bias even for two-step inference, but an improved sharpness when we increase the number of sampling steps to 30. In contrast, FMPE with 30 sampling steps (comparable speed to CMPE) yields a biased posterior. The posterior of FMPE with 1000 inference steps visibly improves but is orders of magnitude slower than CMPE with 30 steps.

## C.7 Experiment 5

**Neural network details.** Both CMPE and FMPE use an MLP with 4 hidden layers of 512 units each and 20% dropout. CMPE and FMPE train for 1000 epochs with a batch size of 64. The consistency model uses $s_0 = 10, s_1 = 50, T_{\max} = 50$. ACF and NSF employ a coupling architecture with learnable permutations, 6 coupling layers of 128 units each, 20% dropout, and 300 epochs of training with a batch size of 64.

The input data consists of multiple time series of different lengths. As proposed by Schmitt et al. [36] for SBI with heterogeneous data sources, we employ a *late fusion* scheme where the input modalities are first processed by individual embedding networks, and then fused. First, the data on tumor size growth curves enters an LSTM which produces a 16-dimensional representation of this modality. Second, the information about radial features is processed by a temporal fusion transformer with 4 attention heads, 64-dimensional keys, 128 units per fully-connected layer, and $10^{-4}$ L2 regularization on both weights and biases. These latent representations are then concatenated and further fed through a fully-connected two-layer perceptron of 256 units each, $10^{-4}$ L2 regularization on weights and biases, and 32 output dimensions.

Figure 7 shows calibration plots for all methods. We observe that our CMPE method with 30 inference steps leads to the best calibration, with affine coupling flows and neural spline flows performing similarly with respect to the expected calibration error (ECE).

**Affine Coupling Flow (ACF)**
RMSE 0.589, max ECE 0.021, **inference time 1.07s**

**Neural Spline Flow (NSF)**
RMSE 0.590, max ECE 0.027, inference time 1.95s

**Flow Matching (FMPE), 30 steps**
RMSE 0.582, **max ECE 0.222**, inference time 17.13s

**Flow Matching (FMPE), 1000 steps**
RMSE 0.583, max ECE 0.057, **inference time 500.90s**

**Consistency Model (CMPE; Ours), 2 steps**
RMSE 0.616, max ECE 0.064, inference time 2.16s

**Consistency Model (CMPE; Ours), 30 steps**
**RMSE 0.577, max ECE 0.018**, inference time 18.33s

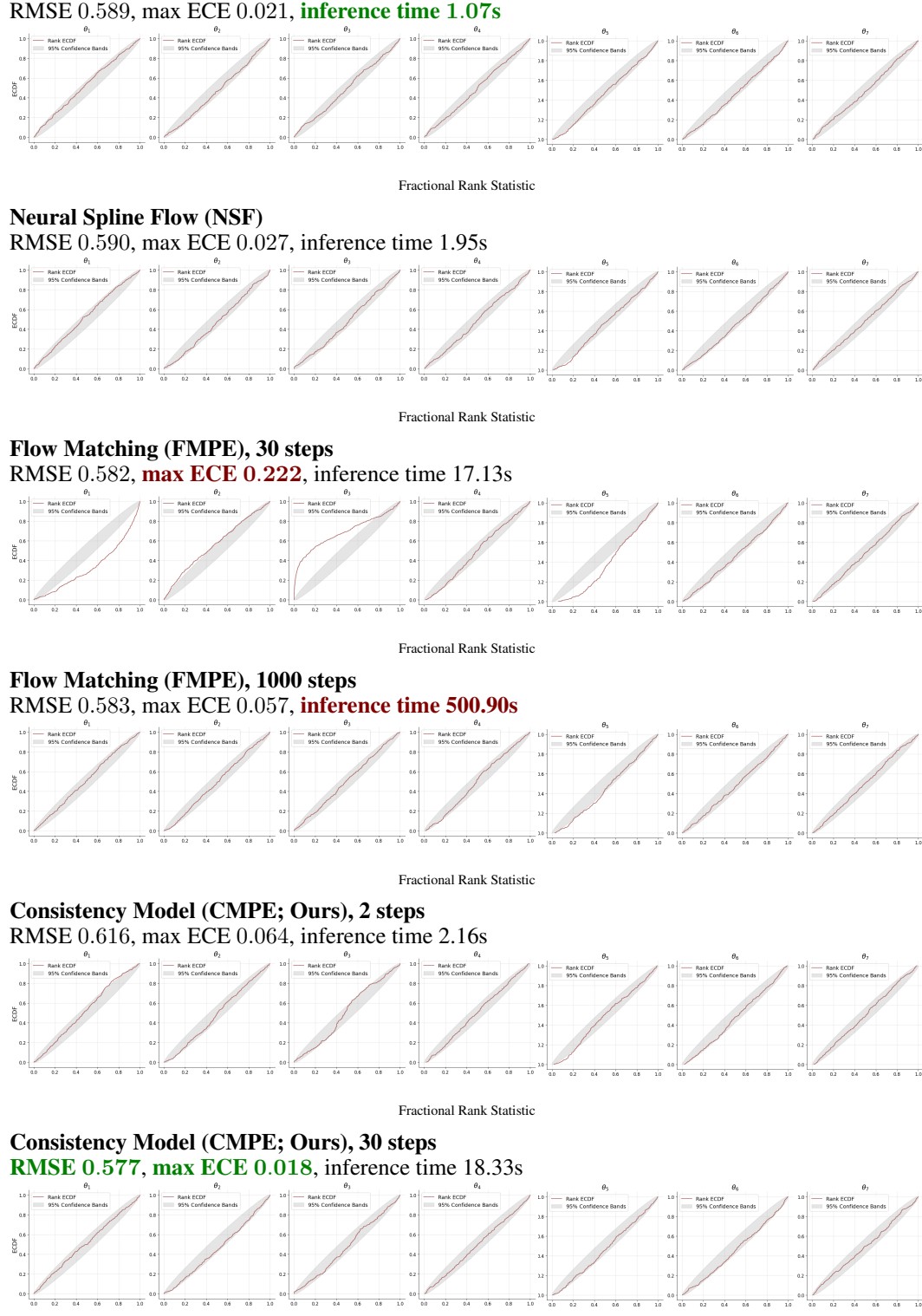

Figure 7: Calibration plots for Experiment 5. The gray envelopes represent the 95% confidence bands for sufficient calibration. Inference times refers to the wall-clock time (in seconds) required to draw 2000 posterior samples. We emphasize the best performance for each metric in green, and the worst performance for each metric in darkred.

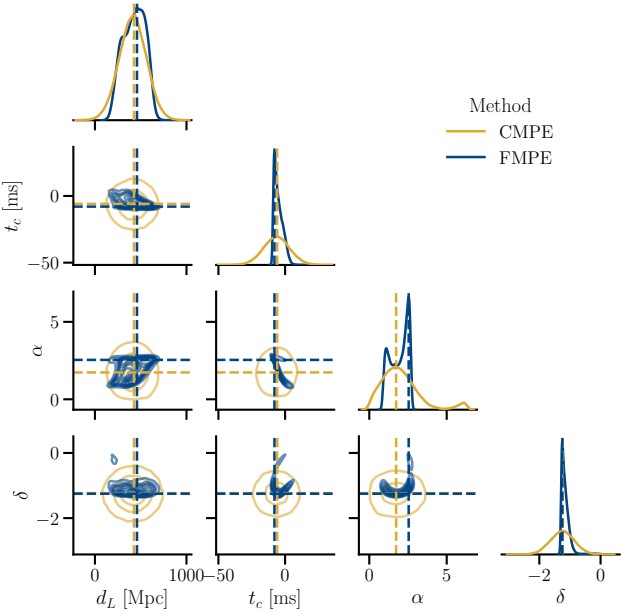

Figure 8: Pairplot of univariate and bivariate posterior draws obtained from CMPE (ours) and FMPE. FMPE clearly outperforms CMPE in this application, where the latter shows pathologically underexpressive posteriors that do not capture the nuances with acceptable detail (see text for details and hypotheses).

## C.8 Experiment 6

In this additional experiment, we apply CMPE to gravitational wave inference of the binary black hole merger GW150914. Our analysis parallels the study of Wildberger et al. [11] who apply FMPE to this challenging scientific task. To this end, we implement our CMPE algorithm in the Dingo library for analyzing gravitational wave data using neural posterior estimation [67]. Dingo implements a sophisticated analysis pipeline that is carefully tailored to gravitational wave inference. As in previous experiments, we compare FMPE and CMPE based on identical neural network backbones with a total of $\approx 190$ Million neural network weights, and we refer to Wildberger et al. [11] for a detailed description of the experimental setup. The consistency model uses $s_0 = 10, s_1 = 1280, T_{\max} = 50, \sigma_{\text{data}} = 1$.

Similar to Wildberger et al. [11], we show the bivariate posterior marginals for a subset of all 14 inference parameters in Figure 8. FMPE successfully captures the posterior geometry [which matches a ground-truth posterior from a slow reference method; see 11]. In contrast, CMPE yields a pathologically underexpressive posterior which mostly resembles a multivariate Gaussian (with some minor mixture contributions for $\alpha$). Yet, the maximum a posteriori (MAP) estimates of CMPE and FMPE mostly align (up to CMPE covering both posterior modes for $\alpha$), which indicates that CMPE does distill information from the model. We suspect that the pathologically poor performance of CMPE is related to (i) issues in our re-implementation of CMPE interfacing the Dingo library; or (ii) misaligned hyperparameters for this specific application. We will continue troubleshooting and invite the interested reader to watch the accompanying GitHub repository for updates.

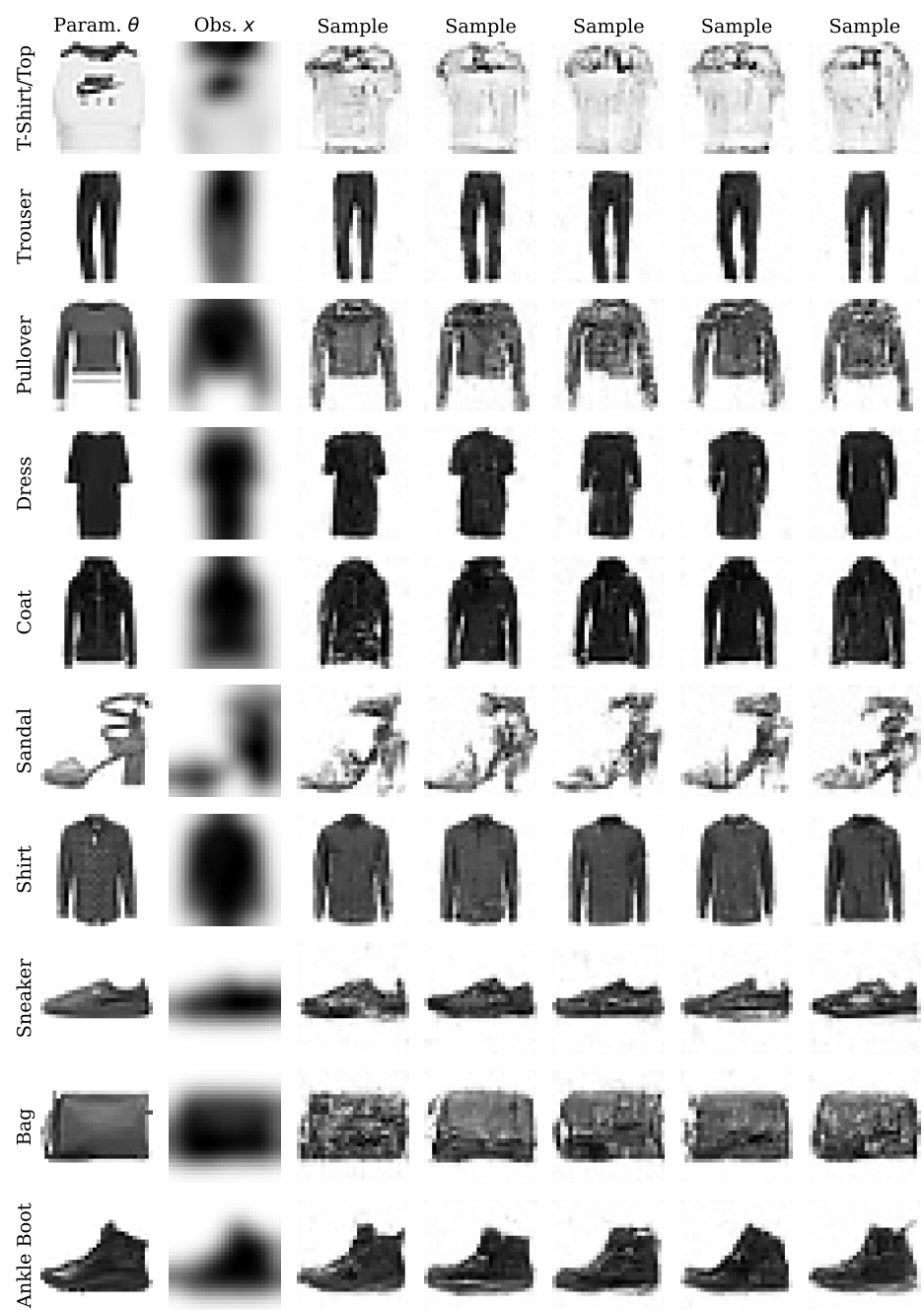

Figure 9: **CMPE - U-Net architecture - 2 000 training images**. CMPE denoising results from each class of Fashion MNIST obtained using a U-Net architecture and two-step sampling. A small training set of 2 000 images was used.

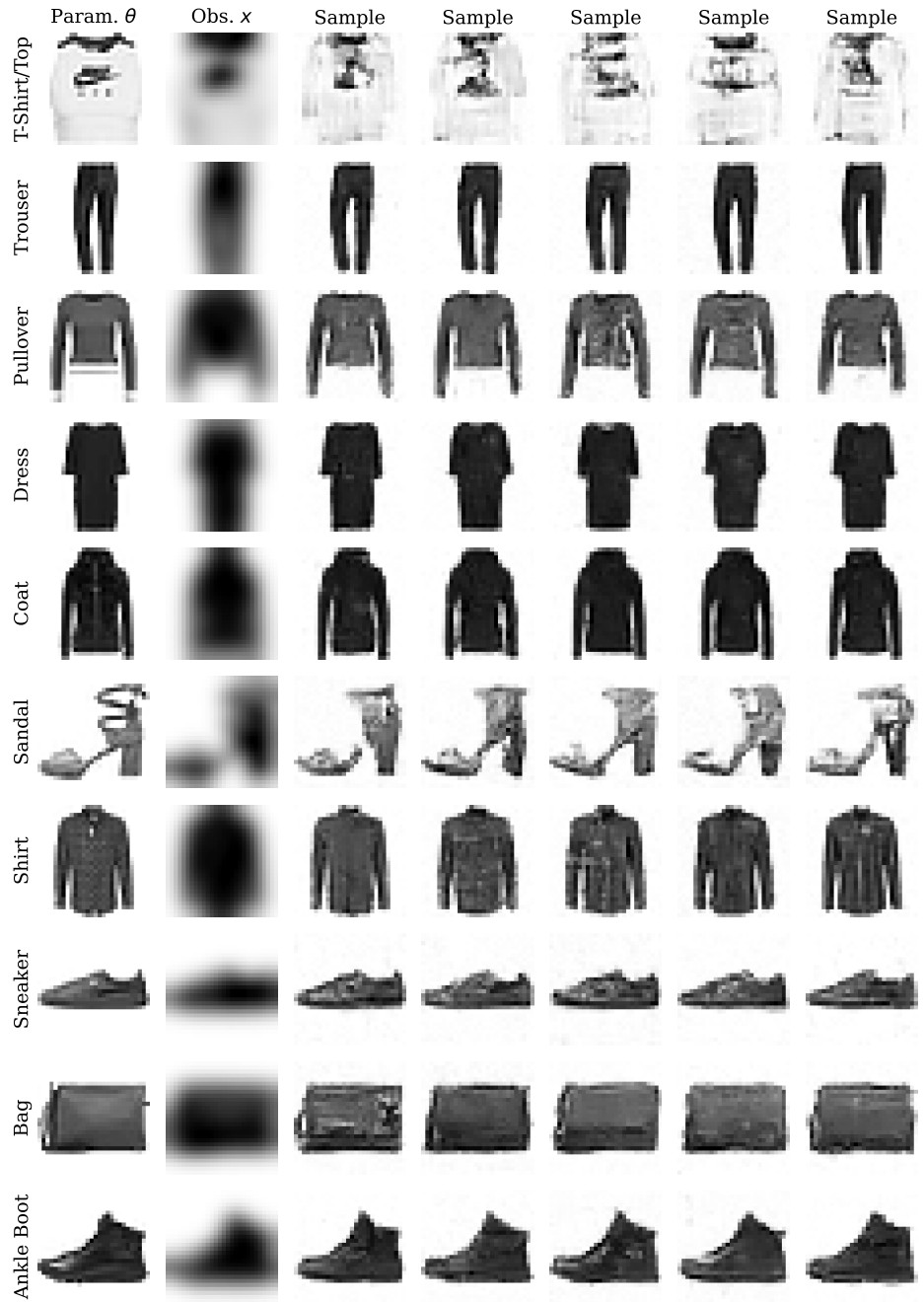

Figure 10: **CMPE - U-Net architecture - 60 000 training images**. CMPE denoising results from each class of Fashion MNIST obtained using a U-Net architecture and two-step sampling. A large training set of 60 000 images was used.

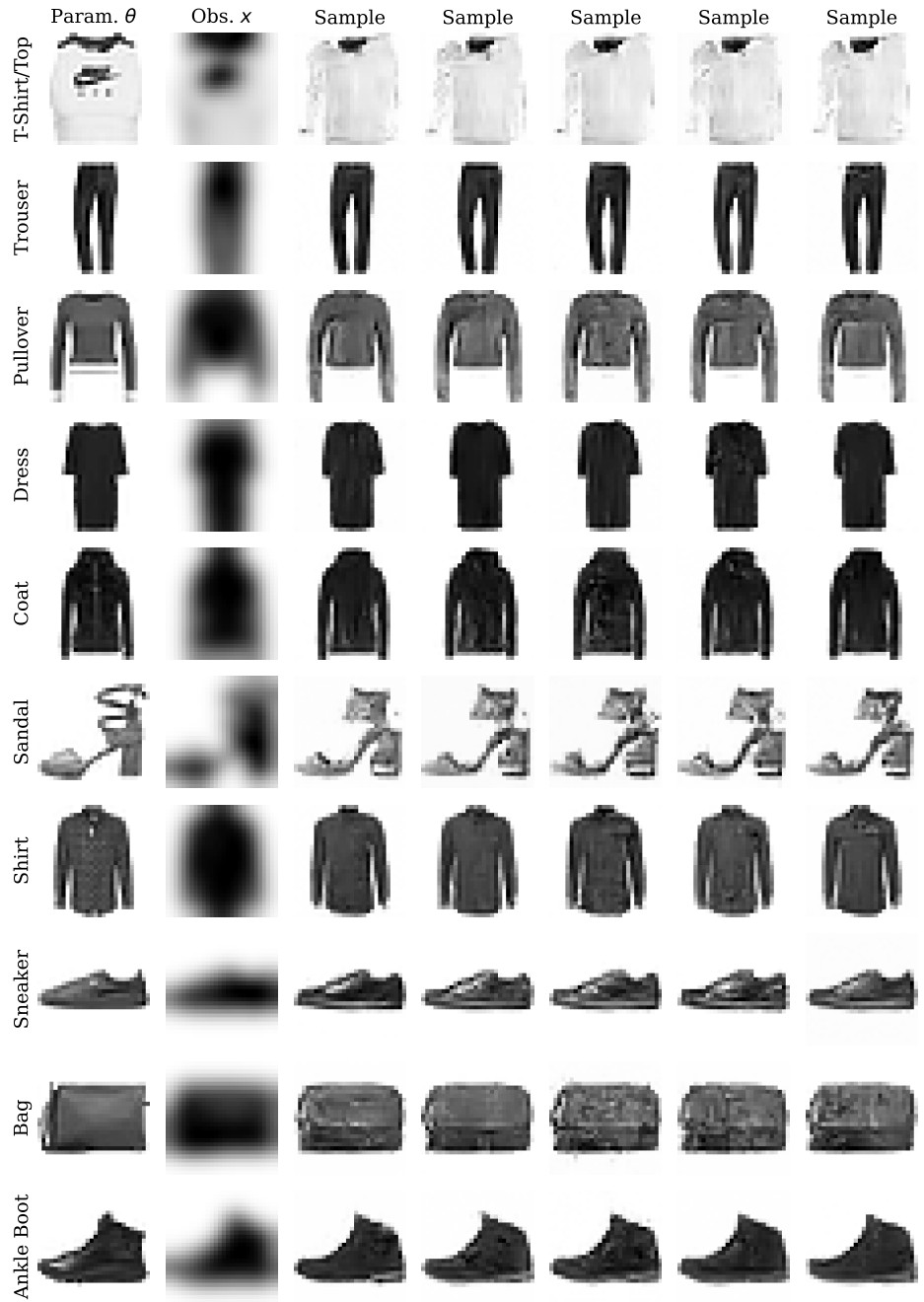

Figure 11: **FMPE - U-Net architecture - 2 000 training images**. FMPE denoising results from each class of Fashion MNIST obtained using a U-Net architecture. A small training set of 2 000 images was used.

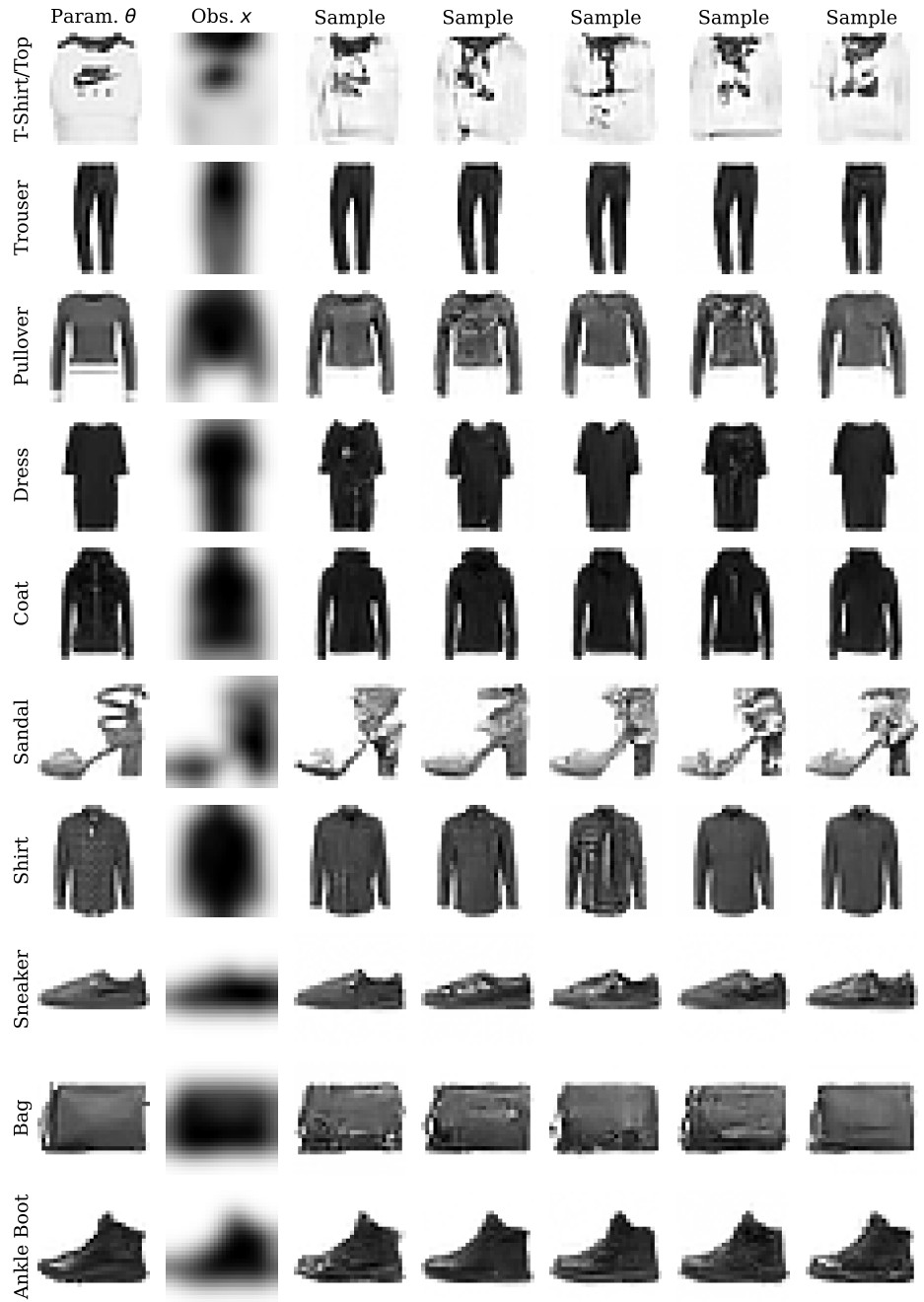

Figure 12: **FMPE - U-Net architecture - 60 000 training images**. FMPE denoising results from each class of Fashion MNIST obtained using a U-Net architecture. A large training set of 60 000 images was used.

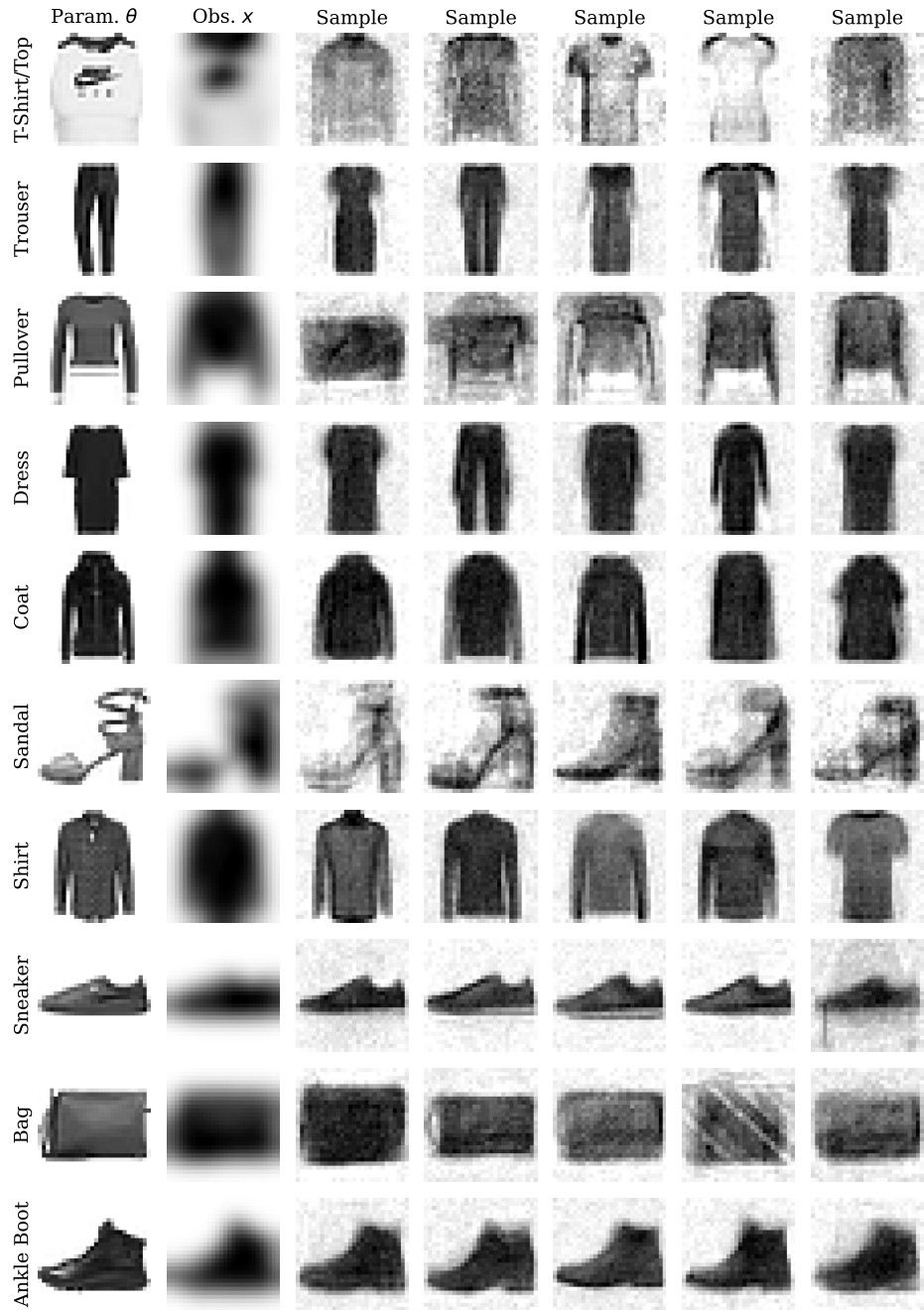

Figure 13: **CMPE - Naïve architecture - 2 000 training images**. CMPE denoising results from each class of Fashion MNIST obtained using the naïve architecture described above and two-step sampling. A small training set of 2 000 images was used.

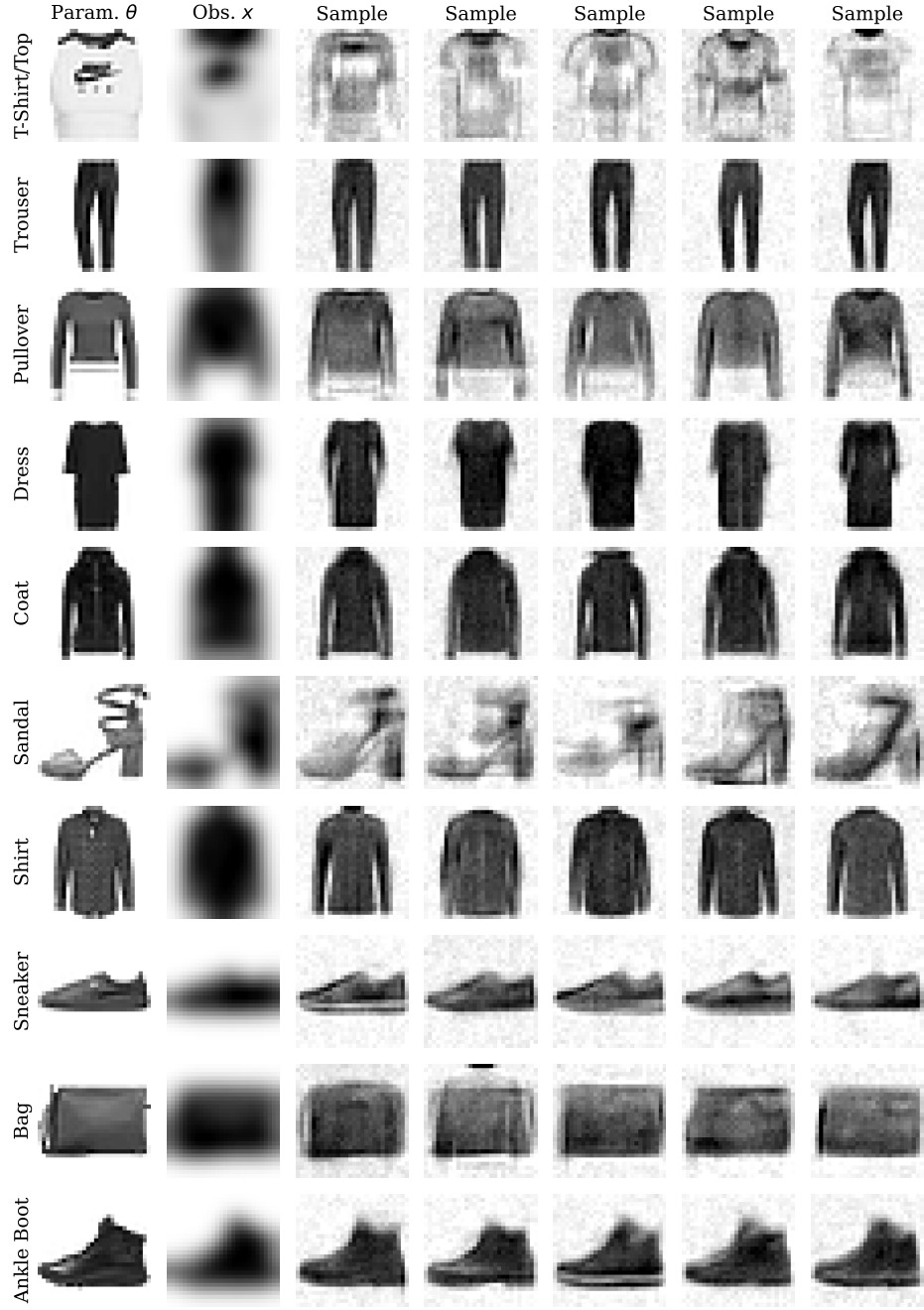

Figure 14: **CMPE - Naïve architecture - 60 000 training images**. CMPE denoising results from each class of Fashion MNIST obtained using the naïve architecture described above and two-step sampling. A large training set of 60 000 images was used.

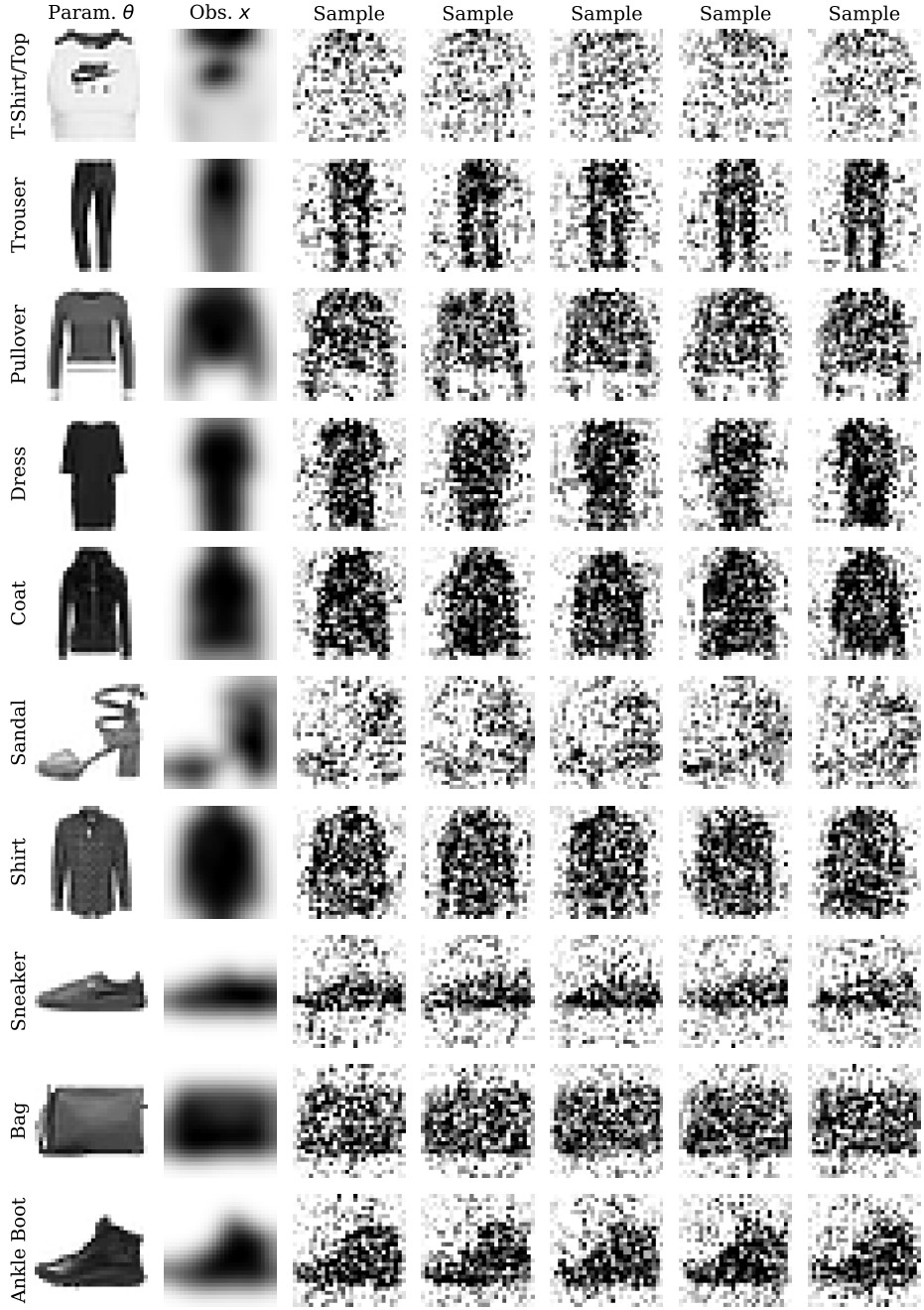

Figure 15: **FMPE - Naïve architecture - 2 000 training images**. FMPE denoising results from each class of Fashion MNIST obtained using the naïve architecture described above. A small training set of 2 000 images was used.

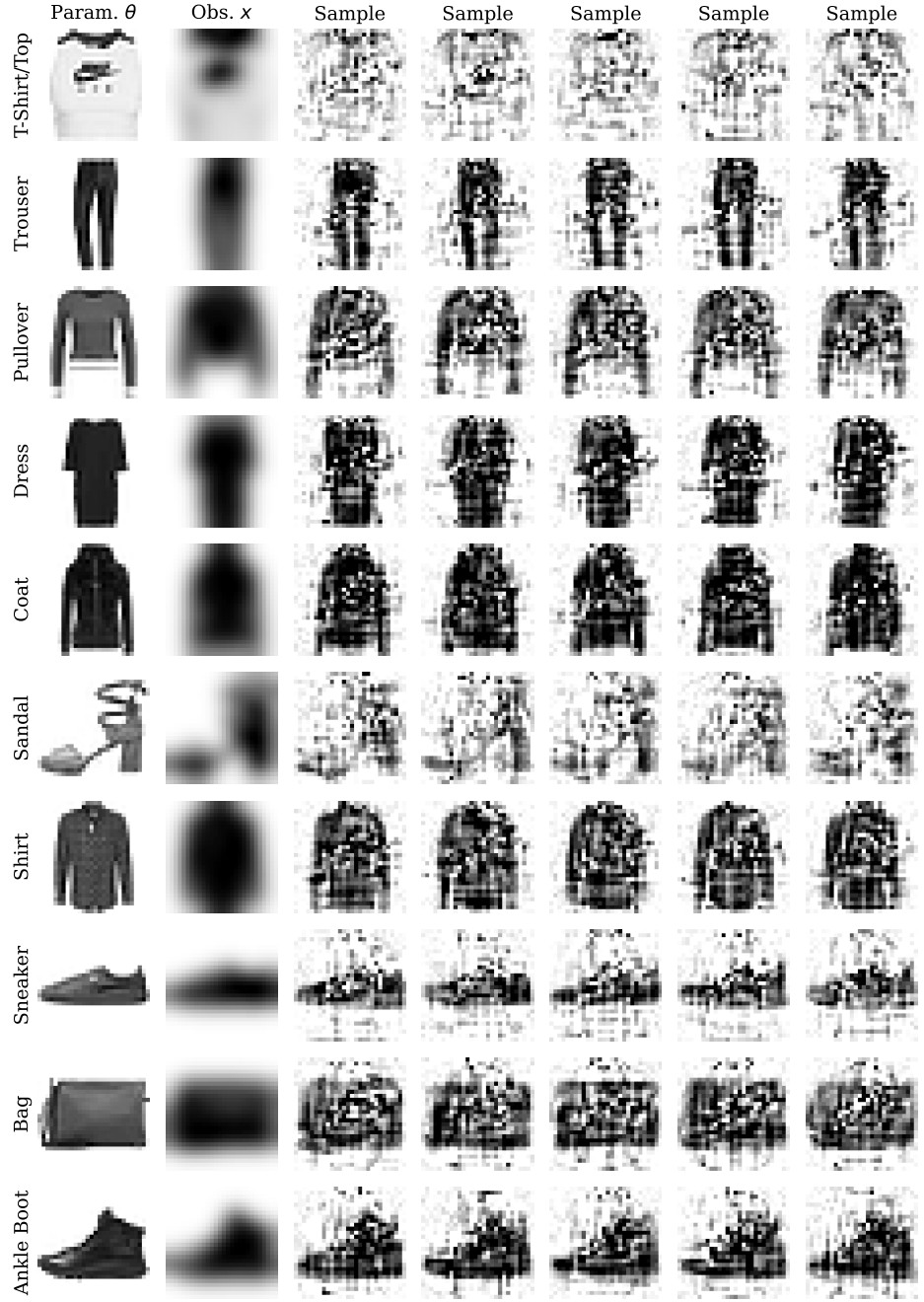

Figure 16: **FMPE - Naïve architecture - 60 000 training images**. CMPE denoising results from each class of Fashion MNIST obtained using the naïve architecture described above. A large training set of 60 000 images was used.

