# OpenReview forum: "Consistency Models for Scalable and Fast Simulation-Based Inference"
_NeurIPS.cc/2024/Conference — NeurIPS 2024 poster_

### Official Review · Reviewer_iNhZ · 2024-06-26

**Soundness:** 3
**Presentation:** 3
**Contribution:** 2
**Rating:** 5
**Confidence:** 4

**Summary:**

The paper adopts consistency models for simulator-based inference tasks, highlighting its expressive free-form architectures and fast inference as main advantages. The new method, called consistency model posterior estimation, was shown to outperform Neural Posterior Estimation (NPE) and be competitive with Fully Neural Posterior Estimation (FMPE) methods in terms of amortized inference quality.

**Strengths:**

The paper demonstrates a thorough experimental analysis of the method, with diverse case studies and a comprehensive list of comparison methods. This suggests that the proposed method is reasonable and applicable for simulator-based inference. The method appears to be novel and addresses a well-established task, improving the speed of inference and simplifying hyper-parameter selection. The submission is mostly technically sound, with the main claims supported. The authors are transparent about the limitations of their work and have published the code, which is commendable.

**Weaknesses:**

The main weaknesses lie in the presentation and the overall results of the paper. The paper is quite difficult to follow, and the overall contribution seems relatively small. The adoption of the consistency model for simulator-based inference (SBI) appears straightforward, and some methodological decisions, such as those in Section 3.3, lack justification. The significance of the contribution may be understated due to the presentation issues in Sections 1-3, which need significant improvement to strengthen the proposal for this new method. Detailed suggestions for improvement are provided below.

**Questions:**

Abstract

(line 1): Consider revising the first sentence "Simulation-based inference (SBI) is constantly in search of more expressive..." to be more concrete.

(line 5): Clarify what is meant by 'free-form architectures'.

(line 5-6): Some SBI methods have long inference times, and 'overcoming sampling inefficiency at inference time' may be bad for them --perhaps some clarification is needed.

(line 8-9): 'Providing an architecture' slightly contrasts with the advantage of having a free-form architecture, so some justification is need.

(line 10): Could you specify the dimensions this line refers to and explain why low-dimensional simulator parameters were difficult to handle with unconditional consistency models and why not simply use other SBI methods instead?

Introduction

(line 20): Could you rephrase to avoid broad generalizations about SBI research, as there are many recent works focused, for instance, on choosing summary statistics with little relation to generative modeling?

(line 31): Could you clarify the term 'free-form architecture'?

(line 36): Please define 'target spaces' in the context of SBI applications.

(lines 41-43): I don't understand the significance of this sentence. Do authors imply that they explore empirically the quality of conditional CMs and it is one of the contributions? It follows the sentence about the empirical evaluations in the paper, so I am a little bit confused.

Section 2

(line 59): I am not familiar with the term 'simulation-based training'. Perhaps the point of the term is to emphasize the synthetic nature of the data generated by the simulator or some other reason?

(line 66, 70): I am not sure why simulation models are presented as 'programs'. This may be confusing as it is not common for the SBI literature in general. Later authors use the term 'program states', but it has very little to do with the setting they operate in. Therefore I would propose to either avoid such terminology or explicate the setting they operate in (if it differs from the 'classical' SBI setting).

Typos: (line 71): 'g( \cdot, \cdot)'. (line 74): 'reasoning' (line 79): 'sequential and amortized'. (line 95): 'it' (line 124): 'they'

(line 101): Please provide a reference for 'optimal transport'.

Section 3

(line 151): x is presented as “a fixed conditioning variable x”, but it is already used for observations in the SBI context. Could you clarify in the text if they are the same?

(line 177): Could you at least briefly detail the ‘hardships for computing the posterior density’ in the main text?

(Eq 8): Could you provide justification or references for this objective?

Experiments

(line 234): Could you provide a reference, where the hyper-parameters for the comparison methods are listed?

**Limitations:**

The authors have adequately addressed the limitations of their work in a separate section.

---

> ### Author Rebuttal · Authors · 2024-08-07
>
> Thank you for your assessment of our work, as well as the detailed list of questions and comments. We responded to your general points below and incorporated all items of your list of questions/edits in our manuscript. We hope that this addresses your concerns regarding the paper’s presentation and justification.
>
> If any remaining reservations prevent you from recommending our paper for acceptance, please let us know and we are happy to further engage in a conversation during the author-reviewer discussion period.
>
> ---
>
> ## Clarifying question to the reviewer
>
> In your summary, you write that our CMPE method is “competitive with Fully Neural Posterior Estimation (FMPE)”.
> Are you referring to Flow Matching Posterior Estimation (FMPE), and “Fully Neural Posterior Estimation” is a typo?
> We observe that our CMPE method is consistently 50–100x faster than FMPE, performs comparably with FMPE in 2 of 5 experiments (w.r.t. accuracy, calibration, or distance to a ground-truth posterior), and even outperforms FMPE in the other 3 experiments. Hence, the conclusion “competitive with FMPE” is an understatement in light of the empirical results we present.
>
> ---
>
> ## W1: Presentation
> > The main weaknesses lie in the presentation and the overall results of the paper. The paper is quite difficult to follow [...]. The significance of the contribution may be understated due to the presentation issues in Sections 1-3, which need significant improvement to strengthen the proposal for this new method. Detailed suggestions for improvement are provided below.
>
> Thank you for voicing concerns about the presentation of our paper. The other reviewers remark that the paper may have a large significance for the SBI community. Therefore, we are particularly interested in making the paper accessible to a broad audience and will gladly make the necessary edits to ensure this.
>
> Within the scope of this review process, we are surprised to see the stark contrast to the assessment of the other three reviewers, who explicitly remark that the paper is well-written (scores: good–good–excellent) and technically sound (scores: excellent–fair–excellent).  We are keen to improve the accessibility of the paper. To this end, we have addressed all of your detailed comments below and updated the manuscript accordingly.
>
> ---
>
> ## W2: Justification of decisions
> > The adoption of the consistency model for simulator-based inference (SBI) appears straightforward, and some decisions, such as those in Section 3.3, lack justification.
>
> Thank you for this observation. Given that consistency models are relatively new, there is no theory (yet) that allows detailed reasoning about their hyper-parameters. Therefore, the decisions were made based on empirical observations. Most of the design decisions were made identical to those presented in [41], with some changes to fit the SBI settings. We will use the additional page of an eventual camera-ready version to add a section that discusses hyper-parameters and explicitly states our recommendations for SBI.
>
> ---
>
> ## Answers to the detailed list as "Official Comment"
>
> Due to space constraints, we have to post our answers to your detailed list of comments and edits as an **Official Comment** object in OpenReview. Please kindly let us know if that is not visible to you.

---

> ### Author Response · Authors · 2024-08-07
> **Response to detailed list (Part 1)**
>
> > (line 1): Consider revising the first sentence "Simulation-based inference (SBI) is constantly in search of more expressive..." to be more concrete.
>
> Thank you, we edited the sentence to:
>
> *Research in simulation-based inference (SBI) aims to develop algorithms that can accurately estimate the unknown parameters of complex simulation models while minimizing computational time and the amount of required training data.*
>
> ---
>
> > (line 5): Clarify what is meant by 'free-form architectures'.
>
> **TL;DR:** We will change the term free-form to unconstrained.
>
> **Detailed answer:** The term “free-form architecture” refers to an architecture of a generative neural network that is not subject to specific design constraints. We use this term following Draxler et al. (2024; [1]). Dax et al. (2023, page 1; [2]) call this “unconstrained architecture”, which is synonymous. For example, most normalizing flow architectures (affine coupling flow, neural spline flow) require a specialized layout that allows cheap computation of the Jacobian determinant, which restricts the design space of the neural networks. Hence, normalizing flows typically don’t allow free-form (unconstrained) architectures.  In contrast, flow matching and consistency models do not require a special neural network architecture, which means that they allow “free-form architectures” that can be more expressive and tailored to the specific data at hand [1]. We acknowledge that the term “free-form” is currently less widespread and will instead use “unconstrained” in the manuscript.
>
> [1] Draxler et al. (2024). Free-form Flows: Make Any Architecture a Normalizing Flow. https://arxiv.org/abs/2310.16624
>
> [2] Dax et al. (2023). Flow Matching for Scalable Simulation-Based Inference. https://arxiv.org/abs/2305.17161
>
> ---
>
> > (line 5-6): Some SBI methods have long inference times, and 'overcoming sampling inefficiency at inference time' may be bad for them --perhaps some clarification is needed.
>
> Could you please clarify what you mean by this remark? We argue that SBI methods with long inference times are generally not desirable if we can achieve equal performance with shorter inference times. For example, we empirically show that flow matching (FMPE) is accurate but slow, and consistency models are as accurate (or more accurate depending on the task) but up to 100x faster.
>
> ---
>
> > (line 8-9): 'Providing an architecture' slightly contrasts with the advantage of having a free-form architecture, so some justification is need.
>
> While consistency models allow for unconstrained free-form architectures, we would like to provide readers and analysts with sensible default architectures as a starting point. We will clarify this sentence in the abstract.
>
> ---
>
> > (line 10): Could you specify the dimensions this line refers to and explain why low-dimensional simulator parameters were difficult to handle with unconditional consistency models [...]?
>
> This line refers to the dimensionality of the parameter space in SBI. In the public review/rebuttal by Song et al. (2023, [1]), the authors report that consistency models are particularly efficient in high-dimensional applications and are expected to struggle in low-dimensional tasks. However, they do not give a theoretical reason for this. In our work, we mitigate this shortcoming through a revised set of default hyperparameters which led to excellent performance in a range of SBI tasks, and this is one contribution of our paper.
>
> [1] https://openreview.net/forum?id=WNzy9bRDvG&noteId=EQQd87ImQk
>
> > [continued] …  and why not simply use other SBI methods instead?
>
> Our comparisons feature multiple state-of-the-art SBI methods (NPE with affine coupling flows, NPE with neural spline flows, FMPE), and we show that consistency models outperform these other SBI methods in almost all cases by achieving fast and accurate inference.
>
> ---
>
> > (line 20): Could you rephrase to avoid broad generalizations about SBI research, as there are many recent works focused, for instance, on choosing summary statistics with little relation to generative modeling?
>
> Thank you, we agree with you and propose the following clarification:
>
> *Recently, multiple streams of neural SBI research have been capitalizing on the rapid progress in generative modeling of unstructured data by re-purposing existing generative architectures into general inverse problem solvers for applications in the sciences [7–10].*
>
> ---
>
> > (line 31): Could you clarify the term 'free-form architecture'?
>
> We changed the term to “unconstrained”. See your comment and our response above for more context.
>
> ---
>
> > (line 36): Please define 'target spaces' in the context of SBI applications.
>
> Thanks, we will replace ‘target spaces’ with ‘parameter spaces’ to match the standard SBI terminology. If you prefer another expression, please let us know and we will be happy to adjust it.

---

> ### Author Response · Authors · 2024-08-07
> **Response to detailed list (Part 2)**
>
> > (lines 41-43): I don't understand the significance of this sentence. Do authors imply that they explore empirically the quality of conditional CMs and it is one of the contributions? It follows the sentence about the empirical evaluations in the paper, so I am a little bit confused.
>
> Yes, it is one of the contributions that we are the first to explicitly study the uncertainty quantification of consistency models (e.g., via simulation-based calibration), which is typically not a concern in the literature on generative image modeling. We agree that the ordering of the sentences is suboptimal here. We will swap the sentences to:
>
> Additionally, the quality of conditional CMs as Bayesian samplers has not yet been explored empirically (e.g., in terms of probabilistic calibration and precision), even though this is crucial for their application in science and engineering. Lastly, while CMs for image generation are trained on enormous amounts of data, training data are typically scarce in SBI applications. In our empirical evaluations, we demonstrate that CMs are competitive with state-of-the-art SBI algorithms in low-data regimes using our adjusted settings (see Appendix A for details).
>
> ---
>
> > (line 59): I am not familiar with the term 'simulation-based training'. Perhaps the point of the term is to emphasize the synthetic nature of the data generated by the simulator or some other reason?
>
> Correct, the training scheme is based on synthetic simulations by the simulator. We change the sentence to:
>
> *In the following, the neural network training relies on a synthetic training set [...]*
>
> ---
>
> > (line 66, 70): I am not sure why simulation models are presented as 'programs'. [...]
>
> **For context:** Presenting simulation models as programs with latent program states dates back to the seminal paper “The frontier of simulation-based inference” [1] and it is equivalent to the classical SBI setting. We agree with your concern and will remove the ‘program’ notion of simulation models in the manuscript to avoid confusion.
>
> [1] https://www.pnas.org/doi/10.1073/pnas.1912789117
>
> ---
>
> > Typos: (line 71): 'g( \cdot, \cdot)'. (line 74): 'reasoning' (line 79): 'sequential and amortized'. (line 95): 'it' (line 124): 'they'
>
> Fixed, thank you.
>
> ---
>
> > (line 101): Please provide a reference for 'optimal transport'.
>
> Thanks, we added [1] and [2] as a reference for optimal transport.
>
> [1] Villani, C. (2009). Optimal Transport. https://link.springer.com/book/10.1007/978-3-540-71050-9
>
> [2] Peyré, G., & Cuturi, M. (2019). Computational Optimal Transport. Foundations and Trends in Machine Learning, 11(5–6), 355–607.https://arxiv.org/abs/1803.00567
>
> ---
> > (line 151): x is presented as “a fixed conditioning variable x”, but it is already used for observations in the SBI context. Could you clarify in the text if they are the same?
>
> In SBI, the observations $x$ are in fact the (fixed) conditioning variables for the neural density estimator. Edit to the text:
>
> *[...] given a fixed conditioning variable (i.e., observation) $\mathbf{x}$.*
>
> ---
> > (line 177): Could you at least briefly detail the ‘hardships for computing the posterior density’ in the main text?
>
> Thanks, edited to:
> *Currently, this comes at the cost of explicit invertibility, which limits the computation of posterior densities. More precisely, single-step consistency models do not allow density evaluations at an arbitrary parameter value $\boldsymbol{\theta}$ but only at a set of $S$ approximate posterior draws* $\\{\boldsymbol{\theta}\_{\varepsilon}^\{(1)\}, \ldots, \boldsymbol{\theta}_{\varepsilon}^\{(S)\}\\}$.
> *However, this is sufficient for important downstream tasks like marginal likelihood estimation, importance sampling, or self-consistency losses. In contrast, multi-step consistency sampling defines a Markov chain which cannot be evaluated without an additional density estimator (see Appendix C for details).*
>
> ---
> > (Eq 8): Could you provide justification or references for this objective?
>
> This optimization objective is given in Eq. 5 of Song and Dariwal [1]. The paper was recently published at ICLR, the new reference is given below. We adapted the equation to feature a conditioning variable, as required in SBI, and made minor changes to match our notation. We have included the reference to [1] in the updated version of the manuscript:
>
> *We formulate the consistency training objective for CMPE, which extends the unconditional training objective from Song & Dhariwal (2024) with a conditioning variable $\mathbf{x}$ to cater to the SBI setting,
> `<Equation 8>`,
> where [...]*
>
> [1] Song, Y., & Dhariwal, P. (2024). Improved Techniques for Training Consistency Models. The Twelfth International Conference on Learning Representations. https://openreview.net/forum?id=WNzy9bRDvG
>
> ---
> > (line 234): Could you provide a reference, where the hyper-parameters for the comparison methods are listed?
>
> Thanks, we added:
>
> *Appendix D lists hyperparameters of all models in the experiments.*

---

> > ### Comment · Reviewer_iNhZ · 2024-08-08
> > **Rebuttal Response**
> >
> > I would like to thank the authors for their professional rebuttal. I feel it is necessary to clarify certain critical points raised regarding my critique of the presentation and perceived performance of your method:
> >
> > 1. In my review, I made a concerted effort to point out specific areas of unclarity in the presentation of your method, particularly in critical sections of the text. I explicitly mentioned that these issues could lead to the contribution being understated, which influenced my decision to lower my confidence score.  Therefore, I believe it is fair to mention that the presentation has affected my assessment of your work and to list it as a primary weakness. Presentation is inherently subjective, and discrepancies between reviewers on this point are not uncommon. I encourage you to refer to the detailed comments in my Questions section for precise feedback on the presentation.
> >
> > 2. I acknowledge the mistake in the abbreviation. I correctly referred to FMPE but inaccurately expanded it to "Fully Neural Posterior Estimation" instead of "Flow Matching Posterior Estimation". This was a typo, I stand corrected. I trust that the authors did not imply my misunderstanding of the method by questioning whether I meant a different approach.
> >
> > 3. I think that your method is competitive with FMPE and I don't think it's an understatement. Regarding the speed comparison, you use FMPE with 1000 sampling steps while your method uses 30 sampling steps when claiming that your method is faster. If we consider speed alone, this comparison might seem unfair. Table 2 shows that FMPE with 30 sampling steps is marginally faster than your method in Experiment 5. Additionally, in the same Table, FMPE with 30 sampling steps outperforms FMPE with 1000 sampling steps, suggesting potential overfitting in the latter case. As for accuracy, based on your empirical evaluations, CMPE does not consistently outperform FMPE; it varies depending on the scenario. As mentioned in my review, I believe your method is competitive and reasonable. As you state in line 365, "CMPE emerges as a competitive method for SBI", and I agree with this assessment without intending to understate your work.
> >
> > I hope this clarifies my position. Below are specific responses to the rebuttal:
> >
> > * Regarding the justification of decisions, your response notes that "there is no theory (yet) that allows detailed reasoning about their hyper-parameters" and that "the decisions were made based on empirical observations". However, it still requires some justification in the text. As a reader, I am unable to understand this unless you explicitly point it out. It is perfectly acceptable to adjust design decisions empirically, but this needs to be clearly stated when you introduce those decisions. Others using your method will need guidance or some intuition on how to adjust different components of your method if it fails on some tasks.
> >
> > * Regarding lines 5-6, I am not suggesting that SBI methods with long inference times are desirable. The initial sentence reads, "a new conditional sampler for SBI that ... overcomes their sampling inefficiency at inference time". I interpreted "inference time" as "[simulator-based] inference time". Some SBI methods, such as most ABC methods, may have very slow inference times, potentially taking hours depending on the simulator. Therefore, the statement "overcoming sampling inefficiency at [simulator-based] inference time" might not be advantageous in the general SBI sense -- you clearly meant something different, so I proposed to clarify that.
> >
> > * Regarding response to my question "why not simply use other SBI methods instead?" (line 10), there are simpler SBI methods (e.g., BOLFI, ABC, KDEs) that perform well in lower-dimensional cases. Typically, higher-dimensional cases are problematic for SBI, but there are numerous fast and high-accuracy methods for lower-dimensional cases, which may outperform NPE and FMPE in those scenarios.
> >
> > * Regarding "For context: presenting simulation models as programs with latent program states dates back to the seminal paper 'The frontier of simulation-based inference'", it is worth noting that simulator-based inference, also known as "likelihood-free inference", dates back at least to the 1990s and originates from statistics. Including references to these earlier seminal papers could provide a more comprehensive historical context.

---

> > > ### Author Response · Authors · 2024-08-09
> > >
> > > Thank you for replying to our rebuttal, we appreciate your time and the helpful clarifications you provide.
> > >
> > > We are grateful for your efforts to point out unclarities and thank you for the recommendations you provided. We are keen to improve the paper’s clarity and therefore gladly incorporated all proposed edits in our manuscript. We did not mean to discount your critique of our presentation but merely wanted to point out the heterogeneity in the reviewers’ assessment of that aspect. We apologize if we failed to bring this message across in our initial rebuttal.
> > >
> > > In the following, we would like to respond to your helpful clarifications and answers (we shortened some block quotes due to the character limit in OpenReview).
> > >
> > > ---
> > > > Regarding the speed comparison, you use FMPE with 1000 sampling steps while your method uses 30 sampling steps when claiming that your method is faster. If we consider speed alone, this comparison might seem unfair.
> > >
> > > Because CMPE and FMPE use identical neural networks to ensure a fair comparison, the number of inference steps to achieve good results is the natural hyperparameter to influence the speed-accuracy trade-off of the methods. In Experiments 1-3, the performance of few-step FMPE is unacceptably bad (see for example Figures 1 and 2).
> > > This is one of the main points from the paper: CMPE can reach a performance with few steps that FMPE can only achieve with many steps. Therefore, with CMPE we can get away with fewer inference steps, resulting in increased sampling speed. So what is important is (number of sampling steps needed)x(time per step). As the difference in necessary sampling steps dominates, the difference in the time per sampling step is not as important here.
> > >
> > > ---
> > > > Table 2 shows that FMPE with 30 sampling steps is marginally faster than your method in Experiment 5. Additionally, in the same Table, FMPE with 30 sampling steps outperforms FMPE with 1000 sampling steps, suggesting potential overfitting in the latter case
> > >
> > > While the RMSE of FMPE with 30 steps is slightly lower (better) than the RMSE of FMPE with 1000 steps, the calibration is extremely bad, as quantified by max ECE in Table 2 and also in the calibration curves that we provide in the PDF of the general rebuttal (at the top in OpenReview). Therefore, we respectfully disagree with the statement that FMPE with 30 sampling steps outperforms FMPE with 1000 sampling steps when we account for uncertainty quantification.
> > >
> > > ---
> > > > Regarding the justification of decisions [...] this needs to be clearly stated when you introduce those decisions. Others using your method will need guidance or some intuition on how to adjust different components of your method if it fails on some tasks
> > >
> > > We wholeheartedly agree with this and are happy to dedicate more space to the justifications in an eventual camera-ready version which allows one more page.
> > >
> > > ---
> > > > Regarding lines 5-6, [...] Some SBI methods, such as most ABC methods, may have very slow inference times, potentially taking hours depending on the simulator. Therefore, the statement "overcoming sampling inefficiency at [simulator-based] inference time" might not be advantageous in the general SBI sense -- you clearly meant something different, so I proposed to clarify that.
> > >
> > > We indeed misunderstood your initial comment and apologize for the inconvenience. Thank you for the clarification, and we are glad to realize that we are on the same page here.
> > >
> > > ---
> > >
> > > > Regarding response to my question "why not simply use other SBI methods instead?" [...]
> > >
> > > Thank you for this clarification, we misunderstood the initial comment and apologize for that. We will expand the section on SBI methods with those fast-and-high-accuracy methods for lower-dimensional data. We agree that the requirements of a modeler differ depending on the tasks and NPE/FMPE/CMPE are not the right hammer for every nail.
> > >
> > > ---
> > >
> > > > Regarding "For context: presenting simulation models as programs with latent program states dates back to the seminal paper 'The frontier of simulation-based inference'", it is worth noting that simulator-based inference, also known as "likelihood-free inference", dates back at least to the 1990s and originates from statistics. Including references to these earlier seminal papers could provide a more comprehensive historical context.
> > >
> > > Thank you for this clarification and suggestion. We will gladly include references to these earlier seminal papers to paint a comprehensive historical context of the LFI/SBI field.

---

> > > > ### Comment · Reviewer_iNhZ · 2024-08-09
> > > > **Final Rebuttal Response**
> > > >
> > > > I thank the authors for their clarifications. After revisiting the experiments in light of their comments, I recognize that FMPE indeed performs poorly on tasks requiring mode separation. This may be due to suboptimal hyperparameter settings, though as I understand the CMPE hyperparameters were not adjusted either. It only seems fair to conclude that CMPE indeed performs better for multimodal posteriors.
> > > >
> > > > While I still believe comparing 1000 steps of FMPE against 30 steps of CMPE in terms of speed is unfair, I now better understand the authors' point that their method achieves comparable performance in shorter time and may even outperform FMPE in accuracy with more steps. A suggestion for the revised version would be to include all evaluation metrics in one table for easier comparison, though I understand this may be unreasonable to expect in the short term of the discussion period. Overall, the method offers a clear empirical performance gain with about a 30% increase in training time, which seems justified.
> > > >
> > > > I appreciate the significant effort the authors have made to improve the presentation and address my critical points. In fairness, I have adjusted my score accordingly:
> > > >
> > > > (Old score → New score)
> > > > Presentation: 2 → 3
> > > > Rating: 3 → 5
> > > > Confidence: 3 → 4

---

> > > > > ### Author Response · Authors · 2024-08-12
> > > > >
> > > > > We appreciate the constructive discussion and are grateful that you adjusted your score to honor the efforts we've made to address your concerns. We are optimistic that these changes have substantially strengthened the paper. Thank you for your time and service in reviewing our work.

---

### Official Review · Reviewer_bX9J · 2024-07-08

**Soundness:** 4
**Presentation:** 4
**Contribution:** 2
**Rating:** 6
**Confidence:** 4

**Summary:**

The paper proposes to use a conditional consistency model for amortized likelihood-free inference. Empirical evaluation shows that this approach compares favorably in terms of inference time as well as performance against competing methods.

**Strengths:**

- The paper is the first to propose the use of consistency models, which is a fairly new family of models, in the likelihood-free inference task.
- The quality of the experiments conducted and the results obtained is high.
- The paper is well written. The methods used are well described with references to sources.
- The presented results may encourage practitioners to use the described method.
- The authors identify the lack of density evaluation from the consistency model as a weakness of the proposed method, and discuss potential solutions to overcome it.

**Weaknesses:**

- The main weakness of this work is the lack of novelty. It comes down to applying an existing model that shows very good performance in computer vision to likelihood-free inference, which has not been published yet. However, someone has to be first, and this work is well suited to this purpose due to its high quality.
- Little attention is paid to all the "design choices" summarized in Table 3. For comparison, reference [41], on which the authors base their work, presents an analysis of the impact of the hyper-parameters used.
- The authors do not evaluate the proposed solutions to the lack of density evaluation limitation.

**Questions:**

1. Could you provide the coverage curves from which ECE is calculated?
2. Fig 2 b), Fig 4 a) and b) - it looks like the standard deviation around the points is 0. Is it so small that one cannot see it in the plot or is it missing from the figures?
3. Why is ECE reported only for Experiment 5?

**Limitations:**

Yes, the limitations of the proposed method are described.

---

> ### Author Rebuttal · Authors · 2024-08-07
>
> Thank you for your thorough review and the excellent questions you raise. We appreciate that you attest excellent scores regarding presentation as well as soundness, and we are delighted by your optimism that our paper may encourage practitioners to use CMPE in their work.
>
> ---
>
> ## W1: Lack of novelty
> > The main weakness of this work is the lack of novelty. It comes down to applying an existing model that shows very good performance in computer vision to likelihood-free inference, which has not been published yet. However, someone has to be first, and this work is well suited to this purpose due to its high quality.
>
> We appreciate your observation regarding the novelty of our approach. While it is true that our study transfers an existing model to the domain of likelihood-free inference, we believe that continuous exploration and benchmarking of algorithms across different fields are crucial. This is especially true since likelihood-free inference offers a unique test bed for an objective assessment of the statistical performance of conditional generative models and thus informs the mainstream applications about appropriate design choices.
>
> ---
>
> ## W2: Design choices
> > Little attention is paid to all the "design choices" summarized in Table 3. For comparison, reference [41], on which the authors base their work, presents an analysis of the impact of the hyper-parameters used.
>
> Thank you for this remark. We will use the additional page of the camera-ready version to add a section that discusses hyper-parameters and explicitly states our recommendations for SBI. In our experiments, we identified $s_0$, $s_1$, and $T_{\mathrm{max}}$ to be the most relevant to tune in order to achieve good sharpness and calibration. We will highlight this and more prominently refer to reference [41] for a discussion of the remaining hyper-parameters, to keep the paper concise.
>
> ---
>
> ## W3: Density evaluation limitation
> > The authors do not evaluate the proposed solutions to the lack of density evaluation limitation.
>
> You are correct, and we are now more explicit about the density evaluation limitation. While this is not a problem for posterior sampling, it will require more research until consistency models can be used, for example, as surrogate likelihoods in tandem with MCMC samplers.
>
> ---
>
> ## Q1: ECE coverage plots
> > Could you provide the coverage curves from which ECE is calculated?
>
> Thank you for this excellent suggestion! We computed coverage curves for all methods and compiled them in the PDF that we attached to the “Author Rebuttal” at the top of the OpenReview page (could not attach to this reviewer-specific rebuttal). Further, we will add the coverage curves in the updated paper version.
>
> ---
>
> ## Q2: Uncertainty bands in figures
> > Fig 2 b), Fig 4 a) and b) - it looks like the standard deviation around the points is 0. Is it so small that one cannot see it in the plot or is it missing from the figures?
>
> The uncertainty bars are vanishingly small, which is a consequence of the stable evaluation (low variation across the test data set). We will revise the figure to increase the visibility of the uncertainty bars.
>
> ---
>
> ## Q3: ECE reporting
> > Why is ECE reported only for Experiment 5?
>
> In experiments 1–3, we have access to ground-truth posterior samples and thus report more powerful metrics that quantify the distance between the approximate and ground-truth posteriors (i.e., C2ST and MMD).
>
> In Experiments 4 and 5, we only have ground-truth parameters (no full posteriors) and thus resort to metrics that do not compare against full posteriors. Posterior inference on images is notoriously ill-calibrated for some pixels, hence we do not report ECE in Experiment 4. Here, visual inspection is a powerful tool to compare different methods, which is why we display images.
>
> Experiment 5 is a scientific application where calibration is achievable and of paramount importance. We compute the probabilistic calibration via simulation-based calibration (SBC; [1]). We currently report ECE in addition to RMSE which quantifies bias and variance and will also add the corresponding coverage curves [2, 3] to the paper (see your question above).
>
> [1] Talts et al. (2018. Validating Bayesian Inference Algorithms with Simulation-Based Calibration. https://arxiv.org/abs/1804.06788
>
> [2] Säilynoja et al (2022). Graphical test for discrete uniformity and its applications in goodness-of-fit evaluation and multiple sample comparison. https://doi.org/10.1007/s11222-022-10090-6
>
> [3] Radev et al. (2023). JANA: Jointly amortized neural approximation of complex Bayesian models. https://proceedings.mlr.press/v216/radev23a.html

---

> > ### Comment · Reviewer_bX9J · 2024-08-08
> >
> > I thank the authors for addressing my questions and sharing the full ECE coverage plots. They made me realize that in the absence of the possibility of density evaluation, the reported ECE is based on per-dimension coverage.
> >
> > I keep my rating.

---

### Official Review · Reviewer_GeMw · 2024-07-15

**Soundness:** 2
**Presentation:** 3
**Contribution:** 2
**Rating:** 5
**Confidence:** 4

**Summary:**

Adopting the idea from consistency models in the generative process, the authors propose its application for posterior estimation, introducing a new type of model for simulation-based inference (SBI). The proposed method, consistency models for posterior estimation (CMPE), enjoying the benefits of the consistency models, can support unconstrained architecture and is more efficient in the sampling process. Empirically, the authors conducted experiments on low-dimensional datasets, including Two Moons, GMMs, and Kinematics, and high-dimensional datasets, including Fashion MNIST.

**Strengths:**

Overall, the paper is well-written and easy to follow.

The authors try to address the computational efficiency problem of the current SBI methods, which is essential.

The authors provided code and implementation details of the experiments, implying high reproducibility of their results.

**Weaknesses:**

W1: The current experiments might be based on relatively simple datasets. While these are useful for initial validation, they may not adequately demonstrate the model's ability to handle real-world complexities and high-dimensional data. The authors could consider more complex datasets with complex distributions, like multimodal distributions. GW150914 in https://arxiv.org/pdf/2305.17161 might be a good case to try.

W2: The training time for consistency models might be problematic for scaling to high-dimensional distributions, and its stability has been known as a problem for complex distribution.

W3: My main concern is that the motivation for applying a consistency model for SBI might be weak, as consistency models are known to have stability issues for learning complex distributions. The current experiments only verify the computational advantage in the sampling process. However, the increase in the training time and the instability of training might pose more difficulties for real-world applications.

W4: The performance of the consistency models relies heavily on tuning a series of hyper-parameters, which might be a problem for generalization.

**Questions:**

Would you provide a comparison of the training time? And the performance regarding the hyper-parameters of the CMPE.

---

> ### Author Rebuttal · Authors · 2024-08-07
>
> Thank you for your thorough assessment of our work, and for the actionable issues you pointed out. We appreciate that you found our paper well-written, easy to follow, relevant for the SBI field, and reproducible through the code we shared.
>
> ---
> ## W1: Data set selection
> > W1: The current experiments might be based on relatively simple datasets. [...] The authors could consider more complex datasets with complex distributions, like multimodal distributions. GW150914 might be a good case to try.
>
> We agree with you about the need for adequate empirical analyses to evaluate SBI methods. In fact, all three benchmarks in Experiment 1 have multimodal posterior distributions (the inverse kinematics is multimodal in parameter space and we just plot the posterior predictive in 2D data space which is not multimodal). In the updated manuscript, we emphasize this more clearly.
>
> As you acknowledge in your summary, our Fashion MNIST task is quite high-dimensional (784D parameter and data spaces) for typical SBI applications in science. Other tasks, like GW150914, have high-dimensional observations, but low-dimensional parameters. Based on your recommendation, we will apply our CMPE method to GW150914. Due to the long simulation and pre-processing time to generate the training set, we are unlikely to have results during the discussion period. If you have any aspects that we should account for, please let us know and we'll consider them.
>
> ---
> ## W2/W3: Training time and stability
> > W2: The training time for consistency models might be problematic for scaling to high-dimensional distributions, and its stability has been known as a problem for complex distribution.
>
> Thank you for the good points, which we will pay more attention to in the revised version of the paper. The training time for consistency models seems comparable to that of flow matching. Moreover, while constrained architectures (i.e., normalizing flows) need fewer iterations until convergence, the free-form architecture of consistency models (and flow matching) makes for a much faster neural network pass *within every single iteration*. See our response below regarding the good stability of CMPE in SBI.
>
> > W3: My main concern is that the motivation for applying a consistency model for SBI might be weak, as consistency models are known to have stability issues for learning complex distributions. [...]
>
> In our experiments, we did not observe issues regarding instability or infeasible training times for CMPE (if anything, CMPE appeared more stable than normalizing flows without additional tricks, such as heavy-tailed latent distributions). However, we acknowledge that other studies have found problems for high-dimensional image generation tasks, where training times are a common bottleneck due to the sheer scale of the problem.
>
> In our setting of amortized inference, we are typically not very concerned about slightly longer training times, which are usually in the magnitude of minutes to two-digit hours. Instead, we want to achieve maximally accurate, well-calibrated, and fast *inference*. As we demonstrate in our experiments, CMPE achieves this by uniting fast inference speeds (like normalizing flows) with high accuracies (like flow matching). In Experiment 1, we repeated the neural network training for different simulation budgets and did not observe training runs that went “off-rails” and would indicate stability concerns.
>
> ---
>
> ## W4: Hyperparameters
>
> > W4: The performance of the consistency models relies heavily on tuning a series of hyper-parameters, which might be a problem for generalization.
>
> We agree that consistency models introduce additional hyperparameters compared to flow matching and normalizing flows, with the number of hyperparameters being similar to that of modern score-based diffusion samplers. In the paper, we propose a set of hyperparameters that performed well throughout our experiments, and which we suggest as defaults for SBI tasks. We will use the additional page of an eventual camera-ready version to add a section that discusses hyperparameters and explicitly states our recommendations for SBI.
>
> ---
> ## Q1: Training time
>
> > Would you provide a comparison of the training time?
>
> We observe that the training times of CMPE are of the same magnitude as FMPE. In a rough estimate, all else being equal CMPE tends to be 10-25% slower during training. However, using the same number of epochs is an arbitrary choice, it would be better to compare training times required to reach a certain “quality”. As this is hard to define and would probably be very noisy, we did not perform such an experiment to avoid invalid conclusions.
>
> We did not have the time to repeat the entire experimental suite ad-hoc during the rebuttal period but could recreate approximate training times for the inverse kinematics benchmark based on timestamps of our checkpoint directories. We will repeat the benchmark experiments in due time and add the properly measured times to the paper. M denotes the simulation budget. All times in seconds.
>
> | Model| M=512 | M=1024|M=2048|M=4096|M=8192|
> |-|-|-|-|-|-|
> | AC|—| 67| 112| 217|424|
> | NSF| 61| 96|155| 289|495|
> | CMPE (Ours)|128|219| 428| 789| 1483|
> | FMPE|105|193|360| 629| 1236|
>
> ---
> ## Q2: Hyperparameters
> > And the performance regarding the hyper-parameters of the CMPE.
>
> As there are several interacting hyper-parameters, a quantitative analysis of the performance concerning the different hyper-parameters would be highly expensive and probably hard to interpret. We searched this space in an earlier project stage and found a set of parameters that consistently yield good performance in SBI. We gathered some insights on which hyper-parameters are important and are most likely to influence the results. In our experiments, we identified $s_0$, $s_1$, and $T_{\text{max}}$ as the most relevant for sharpness and calibration. We will add a section on hyper-parameters and our recommendations for SBI in a camera-ready version.

---

> > ### Comment · Reviewer_GeMw · 2024-08-07
> > **Acknowledgement of the rebuttal**
> >
> > I thank the authors for the rebuttal and additional experiments. The newly added experiments partly confirm my concern that the training efficiency of CMPE is adverse compared to the other methods. At the same time, the relative gain in the inference time is not significant. I am inclined to keep my current rating.

---

> > > ### Author Response · Authors · 2024-08-09
> > >
> > > Thank you for your response, we appreciate that you take the time to engage in the discussion period.
> > >
> > > We do not quite understand your verdict. Our timing experiments show that CMPE training takes about as long as FMPE training, but our method has up to 75$\times$ faster inference in Experiments 1-3 and up to 1000$\times$ faster inference in Experiment 4. In Experiments 1-3, our inference takes only slightly longer than NSF and ACF, but has much better accuracy. Thus, our method occupies an interesting location on the Pareto frontier of the different modeling objectives and cannot be dismissed as insignificant.
> > >
> > > Please keep in mind that the training times we reported in the rebuttal are based on approximate timings with a fixed number of epochs (not on some early stopping criteria). This means that the small difference between CMPE and FMPE training times is well within the margin of error of the makeshift timing setup.
> > >
> > > We invite you to reconsider your verdict in the holistic context of all empirical evaluations. Thank you for your time and service in reviewing our work.

---

> ### Comment · Reviewer_GeMw · 2024-08-09
> **Response to the authors**
>
> To clarify my above concern, the results you demonstrate in Experiment 5 are what I worry about. Compared to ACF, the best result you report slightly decreases the RMSE from 0.589 to 0.577, while your inference time increases from 1.07s to 18.33s, and your training time increases from 424 seconds to 1236 seconds. Since you emphasize that the goal of the consistency model is to improve inference efficiency while maintaining high accuracy, I find this experiment does not support your argument.
>
> While I am trying to evaluate your results in a holistic way, this result as a big portion of your experiments makes me concerned. I would appreciate if you can help me further address this concern.

---

> > ### Comment · Reviewer_GeMw · 2024-08-12
> > **Response to rebuttal**
> >
> > Dear authors, I notice that we are closing to the end of the discussion period. I am hoping to let you know that I am still waiting for your response for adjusting my decision.

---

> > > ### Author Response · Authors · 2024-08-12
> > >
> > > Thank you for your clarifications and for explicitly encouraging a constructive discussion. Regarding Experiment 5, we understand your concern and appreciate the opportunity to address it.
> > >
> > > We agree with your assessment of this experiment: Depending on the modeling goals, the choice of neural network architecture is not as unanimous as in the other experiments. If modelers desire maximum performance w.r.t. bias/variance (RMSE) and calibration (ECE), CMPE is the best choice according to the assessed metrics. If the modeler wants to optimize for training and inference time at the cost of a slightly worse RMSE and calibration, affine coupling flows (ACF) are the method of choice. In this experiment, we observe the typical trade-off between speed (ACF) and performance (CMPE; ours). As often in computational methods, we empirically see diminishing returns; the performance advantage of CMPE in this task comes with a substantially longer training and inference time compared to ACF. This might be related to properties of the task, such as the unimodal posterior geometry: ACF is known to be efficient in unimodal posteriors while struggling to fully separate multiple posterior modes (see the ‘bridges’ between modes in Figure 1 for ACF). In the other experiments, CMPE empirically shows a clearer superiority in the performance-vs-time trade-off.
> > >
> > > > [...] this result as a big portion of your experiments [...]
> > >
> > > We acknowledge that this experiment does not paint as clear a picture in favor of CMPE as the other experiments. However, it is important to us to refrain from selective reporting and we chose to include this experiment to show the nuanced modeling choices that researchers face when selecting an inference method in SBI. We will use the additional page in an eventual camera-ready version to discuss this aspect in more detail based on Experiment 5.
> > >
> > > We hope that this additional context addresses your concern and again want to express our appreciation for the active discussion you sparked.

---

> > > > ### Comment · Reviewer_GeMw · 2024-08-14
> > > > **Final feedback**
> > > >
> > > > Dear authors, thank you for your response. It's good to know that my assessment was correct. And I appreciate your merit in not cherry-picking the results. I will raise my score to 5.

---

### Official Review · Reviewer_XVP6 · 2024-08-05

**Soundness:** 4
**Presentation:** 3
**Contribution:** 4
**Rating:** 7
**Confidence:** 4

**Summary:**

This paper adapts the recently introduced Consistency Models of
Song et al to the task of Simulation-based Inference. Compared to
the previous approaches based on flow matching, this technique
exhibits similar or better quality. All while being significantly
faster to sample from with fewer restrictions on the underlying
neural architecture.

**Strengths:**

I am not familiar with any work that quite does this. While there do
not seem to be novel technical challenges required to adapt CMs for
the Simulation-based Inference. This is still the first.

This paper is very well written and organised. The contribution is described clearly and contextualised well
in the related work.

The technical quality of this work is very good. All details are well-explained, and the experiments are
reasonably chosen with bost smaller illustrative toy examples and larger more realistic ones.

The work is significant as the massive performance improvements and flexibility in architecture make
this something I expect many researchers to build upon.

**Weaknesses:**

I do believe this paper would benefit for slightly reworked plots since the main advantage is just how much more efficient of an inference method it is. I already appreciate that wall-times are included in many places, but something like Figure 2 would be even better if that was taken into account.

It appears to be a relatively straightforward application of CMs to SBI.

I do think the paper would be stronger if the specific restrictions on the density estimation were mentioned in the main paper vs Appendix C

This paper would benefit from an additional larger example.

**Questions:**

Why for the Gaussian Mixture Model does the C2ST score go up as
the computational budget is increased? Is this just a property of
the dataset being so small?

Would it be possible to have a plot that shows the evaluations as a
function of wall-time? One of the major advantage of this method is
it can get away with doing significantly less sampling, and it would
be nice to have a plot that showcases that

**Limitations:**

Yes they have been adequately discussed in paper.

---

> ### Author Rebuttal · Authors · 2024-08-07
>
> Thank you for your positive review of our work. We appreciate that you find our paper well-written, clearly organized, of very good technical quality, and significant to the field. We have addressed your concerns and are optimistic that this will substantially increase the quality of our manuscript.
>
> ---
>
> ## W1: Plots
>
> > I do believe this paper would benefit for slightly reworked plots since the main advantage is just how much more efficient of an inference method it is. I already appreciate that wall-times are included in many places, but something like Figure 2 would be even better if that was taken into account.
>
> Thank you, we agree that it is an excellent idea to present wall-clock times wherever possible to underscore the massive improvements in speed and performance through our CMPE method.
>
> **Question:** Could you please clarify where exactly you would like us to take wall times into account in Figure 2? Figure 2a contains the wall times (i.e., sampling speed) on the x-axis and Figure 2b contains the wall times (i.e., sampling speed) as labels for each inference method.
>
> If “sampling speed” is too vague a term, we will happily replace it with a term like “inference wall time” or similar.
>
> ---
>
> ## W2: Straightforward application
>
> > It appears to be a relatively straightforward application of CMs to SBI.
>
> We agree that the application of consistency models to SBI is conceptually straightforward, and remark that many of the recent advancements in neural SBI have been fueled by the general progress of generative neural networks (e.g., normalizing flows, flow matching, score-based diffusion, …). We demonstrate that consistency models lead to further progress, especially to a favorable trade-off between sampling time and quality. We further provide default hyperparameters as a starting point for a wide range of SBI applications, which will help practitioners use consistency models for SBI in real-world analyses. Upon publication, we will release an implementation of our CMPE in an open-source Python package which is already used by practitioners in application domains.
>
> ---
>
> ## W3: Density estimation remarks to the main text
>
> > I do think the paper would be stronger if the specific restrictions on the density estimation were mentioned in the main paper vs Appendix C
>
> We agree and we will use the additional page in the camera-ready version to discuss the main points of the density estimation restrictions in the main text.
>
> ---
>
> ## W4: Additional larger example
>
> > This paper would benefit from an additional larger example.
>
> Thanks for this recommendation. Do you have a specific example in mind that would underscore the benefits of CMPE? Based on the comment of reviewer GeMw, we are currently applying CMPE to gravitational wave inference, where the observations (i.e., conditioning variables) are very high-dimensional. Would this address your comment?
>
> ---
>
> ## Q1: Computational budget and performance
>
> > Why for the Gaussian Mixture Model does the C2ST score go up as the computational budget is increased? Is this just a property of the dataset being so small?
>
> We are not completely sure, but your hypothesis aligns with how we tend to interpret it. In this benchmark, we observe that for an increasing computational budget, there is an increased tendency to produce overconfident posteriors. In Hermans et. al [1], Figure 2, we see that the amount of overconfidence can increase in the low-budget regime (e.g. for Spatial SIR/SNPE), but that this isn’t the usual pattern. As can be seen in that figure, usually, the posteriors become less overconfident with an increased computational budget, as one would assume. Therefore we assume that the C2ST score will decrease again for significantly larger budgets.
>
> [1] Hermans, J., Delaunoy, A., Rozet, F., Wehenkel, A., Begy, V., & Louppe, G. (2022). A Crisis In Simulation-Based Inference? Beware, Your Posterior Approximations Can Be Unfaithful. Transactions on Machine Learning Research. https://openreview.net/forum?id=LHAbHkt6Aq
>
> ---
>
> ## Q2: Evaluations as a function of wall-time
>
> > Would it be possible to have a plot that shows the evaluations as a function of wall-time? One of the major advantage of this method is it can get away with doing significantly less sampling, and it would be nice to have a plot that showcases that
>
> We share your opinion that an evaluation as a function of wall time is important to assess the faster inference speed of CMPE while maintaining high inference quality. We provide such a plot in Figure 2a ($x$ axis: inference wall-time, $y$ axis: posterior quality) for the Gaussian Mixture Model experiment and, following your suggestion, will use the additional page in the camera-ready version to present more plots of this type for other experiments.

---

> > ### Comment · Reviewer_XVP6 · 2024-08-13
> > **RE: Rebuttal**
> >
> > I am satisfied with the author response. I think adding the gravitational wave example would be sufficient as a larger example.
> >
> > It might be worth exploring the relationship between computational budget and performance, but I don't think it's essential for this paper. Since the example is fairly synthetic it can't hurt to see if it behaves better if you just generate more points for the GMM.

---

### Author Rebuttal · Authors · 2024-08-07

We thank all reviewers for their assessment of our work. All reviewers agree that conditional consistency models are a promising technique for amortized simulation-based inference, that the topic is relevant to the NeurIPS community, and that our paper is technically sound.

Below, we give a detailed rebuttal for each reviewer, and we have incorporated all comments into a revised version of the manuscript. Thank you for your time and service in reviewing our work, which substantially strengthened the paper and hopefully renders it a valuable contribution to NeurIPS 2024.

**Note.** We observe a strong discrepancy in the assessment of our presentation and clarity of writing: Reviewers XVP6, GeMw, and bX9J conclude that our paper is well-written and easy to follow. The reviewers express this with explicit comments and high scores for presentation (good–good–excellent). In contrast, reviewer iNhZ lists the presentation as the paper’s major weakness. Based on this discrepancy, we would appreciate it if the AC could provide their opinion in this regard as well. We would like to point out that reviewer iNhZ has kindly provided a detailed list of minor edits and clarifications; we have addressed all of these items and hope that this alleviates the remaining reservations.

---

### Decision · Program_Chairs · 2024-09-25

**Decision:**

Accept (poster)

**Comment:**

The paper proposes to use a conditional consistency model for amortized likelihood-free inference.  Their experimental evaluation shows that the inference speed of this approach is much better that existing methods.  While some reviewers initially has some concerns about the experimental evaluation and the clarity of presentation, their concerns seem to have been adequately addressed in the rebuttal phase.